# HISTOGRAM-CONSTRAINED IMAGE GENERATION

## ABSTRACT

Diffusion models have emerged as a dominant paradigm in generative modeling, enabling high-fidelity sampling from complex data distributions. Despite impressive capabilities, controlling diffusion models to produce outputs aligned with user intent remains an open challenge, especially when balancing global coherence with local precision. Existing control mechanisms vary in the granularity of their conditioning signals. For example, textual prompts guide generation globally through high-level semantics, while ControlNet-like approaches secure precise local structure via dense conditions. In this work, we introduce Histogram-constrained Image Generation (HIG), a novel control mechanism that falls into the middle ground of control granularity. Our framework enforces user-specified distributional constraints (e.g., color histograms or latent token distributions) during the generation process with exact precision. We model such control as an optimal transport (OT) problem and apply explicit guidance transformations during sampling, thereby driving the diffusion trajectory to align with the desired histogram. We demonstrate the versatility of HIG across diverse applications, including constrained generation via color/latent histograms and high-capacity information embedding through histogram-level encoding. Our findings underscore the promise of distributional control, a flexible and interpretable control scheme that is fully compatible with existing control mechanisms, diversifying the hybrid strategies for controllable image generation.

## 1 INTRODUCTION

Diffusion models have emerged as a dominant paradigm in generative modeling, enabling high-fidelity sampling from complex data distributions (Sohl-Dickstein et al., 2015; Ho et al., 2020; Dhariwal & Nichol, 2021; Song et al., 2021). As their capabilities grow, there has been an increasing focus on controllable generation – guiding the sampling process to align with user intent. Extensive work explores this direction across diverse control schemes, including text prompts (Rombach et al., 2022; Podell et al., 2023; Labs, 2023), structural guidance (Zhang et al., 2023a; Li et al., 2025), and instance-level content/style customization (Hu et al., 2022; Ruiz et al., 2023).

Intuitively, controllable generation methods can be sorted along a spectrum defined by the information granularity of their control signals. On one end, high-level control schemes guide generation through coarse-grained signals, such as text prompts (Podell et al., 2023; Esser et al., 2024; Labs, 2023) or content/style LoRAs (Hu et al., 2022; Ruiz et al., 2023). These signals typically encode abstract concepts and grant the diffusion process considerable flexibility to improvise during generation. As a result, they influence the output at a global scale, such as specifying the subjects or artistic styles. On the other end, low-level control schemes like ControlNet (Zhang et al., 2023a; Li et al., 2025) typically condition on dense inputs such as edge maps, depth maps, or pose annotations. These methods provide fine-grained spatial control and are particularly effective at anchoring the generation to a given structure or layout.

In this work, we explore a middle ground on the spectrum. We introduce **H**istogram-constrained **I**mage **G**eneration (**HIG**), a novel control scheme that regulates the distributional properties of generated images. HIG is characterized by three key properties:

1. **Flexibility**: it can enforce distributional consistency across diverse choices, such as pixel color histograms or latent token statistics, enabling a wide range of applications.

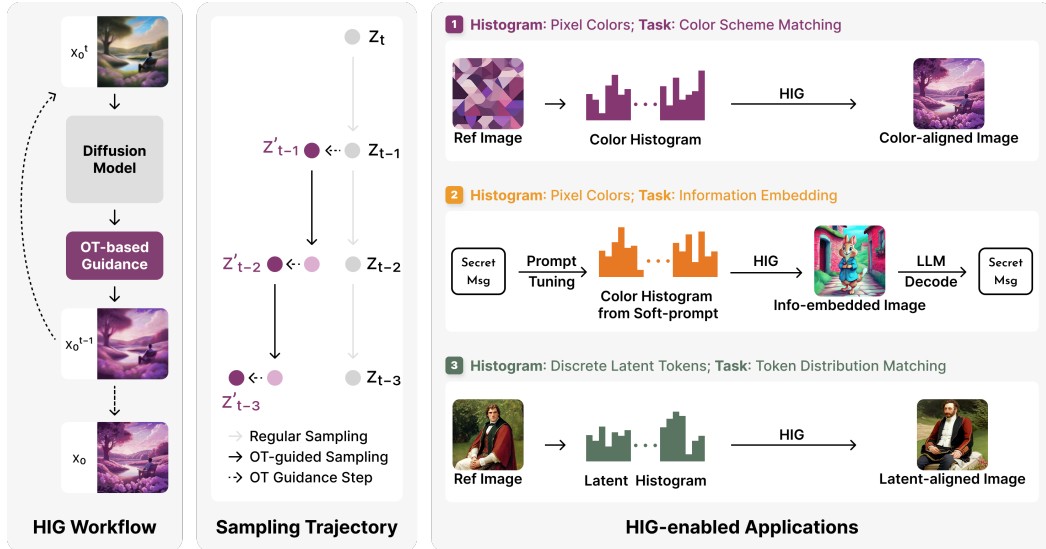

Figure 1: Overview for histogram-constrained generation (HIG). We intervene in the diffusion process with explicit OT-based guidance. HIG enables diverse applications, including constrained generation with arbitrary histogram constraints and high-capacity information embedding.

2. **Precision and Optimality**: the constraint is formalized as an optimal transport (OT) problem (Villani, 2009), which identifies the minimal-cost transformation required to satisfy the constraint exactly (Sec. 3.2). This optimality enables implementing control through a straightforward intervention: directly perturbing the intermediate diffusion outputs. As the perturbation is guaranteed to be minimal in cost, the guidance is mathematically principled and minimally invasive. In addition, the ability to satisfy the constraint precisely opens up applications that demand exact precision (e.g., information embedding).

3. **Lightweight and Compatible Design**: HIG is training-free, and its inference-time overhead is negligible compared to the diffusion generation itself. Unlike most existing control mechanisms that rely on auxiliary inputs or modified architectures, HIG is plug-and-play, interpretable, and compatible with many established methods, such as ControlNets (Zhang et al., 2023a; Li et al., 2025) and LoRAs (Hu et al., 2022; Ruiz et al., 2023).

HIG unlocks a wide range of applications. First, HIG enables generating images that exactly match a given color histogram, offering controls on the global property (overall color schemes) with precision (Sec. 4.1). Second, we demonstrate the control precision of histogram constraints through an information embedding technique, where we embed information via the histogram so that the generated image entails the same information. For example, by deterministically mapping a soft-prompt embedding (Lester et al., 2021) (i.e., a continuous-valued vector) to a target color histogram, our method can encode hundreds of text tokens within a visually indistinguishable image, which can be precisely decoded based on the color histogram (Sec. 4.2). Lastly, we showcase HIG's generalization to latent histograms, where the histograms are computed from the discrete codebooks of image tokenizers (Van Den Oord et al., 2017; Razavi et al., 2019; Esser et al., 2021; Qu et al., 2024; Yu et al., 2024). In such cases, the control effect of latent histograms hinges on the information captured in the latent space, such as low-level colors and textures or high-level semantics. We demonstrate the versatility and control precision of HIG through detailed experiments and analysis.

## 2 INFERENCE-TIME GUIDANCE VIA EXPLICIT TRANSFORMATIONS

Diffusion models are a class of generative models that construct a mapping between a simple noise distribution (typically Gaussian) and complex data distributions via some scheduled interpolations. It involves a forward process that progressively corrupts data with noise, and a reverse process that recovers data from noise. Various mathematical formulations have been proposed to characterize this process, including discrete-time probabilistic models (Ho et al., 2020), continuous-time score-

based SDEs (Song et al., 2020), and deterministic ODEs (Song et al., 2021). Here, we follow the DDPM (Ho et al., 2020) framework to formally describe the proposed inference-time guidance.

The predominant paradigm for high-resolution image synthesis is to train latent diffusion models (LDM) that operate in the latent space of a VAE (Kingma & Welling, 2013) and perform conditional generation in text-to-image fashion (Rombach et al., 2022; Podell et al., 2023). At each sampling step, the LDM takes in the noisy latent $\mathbf{z}_t$ and text condition $\mathbf{c}$ and predicts the quantities that correspond to the training objective (e.g., noise, signal, or velocity), which essentially characterizes the current view of the final image latent $\mathbf{z}_0$. Then, a controlled amount of noise is added back to the intermediate guess $\mathbf{z}_0^t$ to proceed to the less noisy latent $\mathbf{z}_{t-1}$ at the next time step. Formally, at time step $t$, the intermediate guess of a noise-prediction based LDM can be expressed as follows:

$$\mathbf{z}_0^t = \frac{1}{\sqrt{\bar{\alpha}_t}}\mathbf{z}_t - \frac{\sqrt{1 - \bar{\alpha}_t}}{\sqrt{\bar{\alpha}_t}}\epsilon_\theta(\mathbf{z}_t, \mathbf{c}, t), \tag{1}$$

where $\epsilon_\theta(\mathbf{z}_t, \mathbf{c}, t)$ denotes the predicted noise and $\{\bar{\alpha}\}_{t=1:T}$ denote the cumulative noise factors.

We note that most existing control mechanisms are achieved by manipulating $\epsilon_\theta(\mathbf{z}_t, \mathbf{c}, t)$ via auxiliary inputs (ControlNets) or model pieces (LoRAs), thereby deviating from the original trajectory. In contrast, we adopt a complementary approach that explicitly transforms the intermediate prediction $\mathbf{z}_0^t$. Formally, given a guidance transformation $\varphi$ that operate on images, the transformed latent $\mathbf{z}_0^{t'}$ can be obtained through a decode-transform-encode cycle:

$$\mathbf{z}_0^{t'} = \text{VAE-encode}(\varphi(\text{VAE-decode}(\mathbf{z}_0^t))). \tag{2}$$

Hence, we can step to the less noisy latent $\mathbf{z}_{t-1}$ based on the transformed latent:

$$\mathbf{z}_{t-1} = \sqrt{\bar{\alpha}_{t-1}}\mathbf{z}_0^{t'} + \sqrt{1 - \bar{\alpha}_{t-1}}\epsilon_\theta(\mathbf{z}_t, \mathbf{c}, t). \tag{3}$$

Typically, such perturbation only needs to be applied on a subset of the sampling time steps, denoted as $\mathcal{T}$. We note that the OT formulation ensures the intervention is minimally invasive by explicitly minimizing transportation cost. As a result, the transformation remains close to the original diffusion trajectory, allowing high-fidelity images to be generated. In this way, the proposed guidance scheme enables explicit control over the generated image during the de-noising process, while allowing the subsequent diffusion steps to further refine the final output.

## 3 IMAGE GENERATION WITH DISTRIBUTIONAL GUIDANCE

### 3.1 PRELIMINARIES

We define a histogram as a discrete probability distribution over a finite set of $d$ bins, denoted as a vector $\mathbf{h} \in \mathbb{R}^d$ with non-negative entries summing to one. For data with continuous values, such as RGB colors in $[0, 1]^3$, we partition the space into $d$ disjoint bins by uniformly dividing each dimension. For example, a bin may correspond to the colors in $[0, 0.1]^3$, and its representative color can be defined as the mean of each sub-interval. For discrete domains such as VQ-VAE tokens (Van Den Oord et al., 2017), bins naturally correspond to individual token indices, and we can simply count the per-token frequency. Notably, a bin may also correspond to an arbitrary set of tokens/items (e.g., a palette of colors, or a certain subset of VQ codebook tokens), allowing flexible and task-specific binning strategies. For clarity, we term the binning strategy that maps to a single token/item as *single-option binning* and to a subset of tokens/items as *multi-option binning*.

### 3.2 HISTOGRAM MATCHING WITH OPTIMAL TRANSPORT

Given the source image $\mathcal{I}$, we wish to solve for a transformed image $\mathcal{I}' = \varphi(\mathcal{I}, \mathbf{h}^{tgt})$ with minimal content deviation, where $\varphi$ denotes a guidance transformation that matches $\mathcal{I}$ to the target histogram $\mathbf{h}^{tgt}$. To achieve this, we employ optimal transport (OT) (Villani, 2009), a mathematically grounded approach that identifies the most efficient way to transform one distribution into another. In addition to the minimal transformation cost, OT guarantees exact compliance with the target histogram by producing a transport plan that explicitly specifies the quantity of individual units (i.e., number of pixels or tokens) to be reassigned between each source and target bin.

We first discuss the OT formulation for single-option binning, where each histogram bin only consists of a single color range or discrete token. Formally, given the source histogram $\mathbf{h}^{src}$ and the target histogram $\mathbf{h}^{tgt}$, with $\mathbf{h}^{src}, \mathbf{h}^{tgt} \in \mathbb{R}^d$, we want to solve the following optimization problem:

$$\gamma = \arg \min_{\gamma} \langle \gamma, \mathbf{M} \rangle_F, \tag{4}$$

$$\text{s.t. } \gamma \mathbf{1} = \mathbf{h}^{src}, \gamma^{\top} \mathbf{1} = \mathbf{h}^{tgt}, \gamma \geq 0,$$

where $\gamma \in \mathbb{R}^{d \times d}$ denotes the optimal transport plan, $\mathbf{M} \in \mathbb{R}^{d \times d}$ denotes the cost matrix. The cost matrix can be pre-computed based on some distance metric between color tuples or latent embeddings. The OT plan $\gamma$ informs the proportion of every source bin to be transported to a target bin to achieve the desired histogram. We implement the OT transformation by randomly sampling the source pixels/tokens and setting them to the target bin values.

In some cases, strict single-option binning may lead to excessive content distortion during OT-based histogram matching. To mitigate this, we introduce a multi-option binning scheme for OT, where each bin contains multiple candidate values. In this setting, the transport plan only enforces the aggregated mass per bin to match the target histogram, allowing flexible assignments among the intra-bin options to reduce perceptual deviation.

Unlike the single-option setting, the multi-option case requires a more sophisticated formulation due to the added ambiguity within each bin. Formally, let each of the $d$ target bins contain $k$ candidate values, yielding a total of $kd$ options. The OT plan is now defined over a $kd \times d$ matrix, where

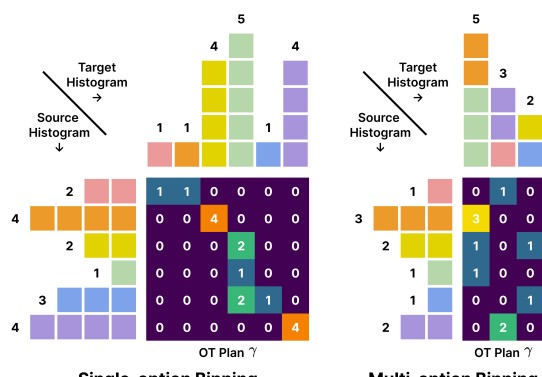

Figure 2: Exemplar OT plans with single-option ($d = 6$) and multi-option binning ($k = 2, d = 3$).

each row corresponds to a specific source option (e.g., a pixel or token value), and each column denotes a target bin. In this case, the target distribution constraint is applied only at the bin level. Within each bin, however, the assignment to specific options remains unconstrained, allowing flexibility in choosing the most perceptually similar mappings. We provide binning illustrations in Fig. 2.

To construct the corresponding cost matrix, we first consider the pairwise distances between all possible options, yielding a cost matrix of size $kd \times kd$. For each entry in the $kd \times d$ OT plan, the cost is defined as the minimum distance from the source option to any of the $k$ options in the target bin. This ensures that, for any source option and target bin, the transport plan always transforms it to the "closest match" in the target bin, reducing unnecessary distortion while still satisfying the bin-level constraints. Formally, we consider $\mathbf{h}^{src} \in \mathbb{R}^{kd}$ and $\mathbf{h}^{tgt} \in \mathbb{R}^d$ in multi-option binning. For the $j$-th option in the $i$-th source bin, its closest match in the $p$-th target bin can be pre-computed and assigned to the cost matrix $\mathbf{M} \in \mathbb{R}^{kd \times d}$, such that:

$$\mathbf{M}_{ik+j,p} = \min_{q} \text{dist}(\mathbf{v}_{ik+j}, \mathbf{v}_{pk+q}), \tag{5}$$

where $i, p \in \{0, 1, \cdots, d-1\}$, $j, q \in \{0, 1, \cdots, k-1\}$, $\mathbf{v}$ corresponds to the stacked option vectors (e.g., $kd$-many color tuples or latent embeddings), $\text{dist}(\cdot)$ denotes a distance metric, such as L1 or L2 distance. When applying the transport plan, we randomly sample a given number of individual units from each source bin and set them to the closest option in the target bin, respectively.

In practice, we use the vanilla network simplex algorithm (Balinski, 1961) to solve for the OT plans. For color histogram matching on $1024^2$ images with $d = 4096$, the optimization takes roughly 0.2 seconds (maxIter = 500K). While faster approximate OT solvers such as Sinkhorn (Cuturi, 2013) (with entropic regularization) are available, the vanilla solver is already good enough for our usage.

## 4 CONSTRUCTION OF TARGET HISTOGRAMS

### 4.1 HISTOGRAMS FROM REFERENCE IMAGES

As a direct approach, we can use histograms extracted from reference images as the target. Specifically, we calculate the histogram of pixel colors or latent tokens and set it as $\mathbf{h}^{tgt}$. During sampling, we apply the guidance transformation $\varphi$ by transporting pixels or tokens according to an optimal transport (OT) plan, resulting in a perturbed output that precisely matches the target histogram.

Notably, we can flexibly adjust the control precision depending on the usage. For color histograms, one may optionally perform a post-hoc OT step on the generated image to enforce exact compliance with $\mathbf{h}^{tgt}$. This guarantees histogram alignment but may introduce visual artifacts (e.g., locally inconsistent pixels), due to the rigid reassignment of colors. Such exact control is necessary in use cases that demand perfect compliance to the histogram constraint (e.g., information embedding), but it can be safely omitted in more relaxed settings (e.g., color scheme matching for aesthetics).

### 4.2 HISTOGRAMS FROM CONTINUOUS VECTORS

In addition to using reference images, we can also construct target histograms from continuous vectors. We showcase this with an *information embedding* technique, whose workflow is illustrated in Fig. 3. Intuitively, the objective is to embed hidden information into an image such that it can be reliably recovered through decoding. In practice, we extend our color-constrained generation framework by replacing the target histogram with a synthetic color distribution derived from a continuous-valued vector (referred to as a soft-prompt embedding). The soft-prompt embedding is optimized to condition an LLM to reproduce a target sentence. Since accurate decoding requires exact alignment between the color histogram and the embedding, this task highlights the strength of our method in enforcing precise constraints through explicit OT-based guidance.

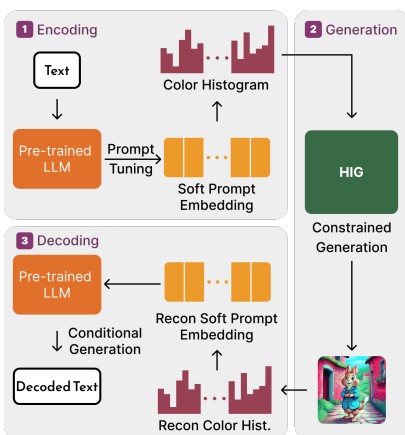

Figure 3: Information embedding workflow.

We first elaborate on how a sequence of text tokens can be transformed into a compact soft-prompt embedding via prompt tuning (Lester et al., 2021). Prompt tuning is a parameter-efficient fine-tuning (PEFT) technique that learns a set of continuous embeddings (soft prompts) that are prepended to the input text to guide the language model's behavior, which can be viewed as task-specific instructions in the embedding space. In our case, we need a soft prompt that can condition the LLM to output an exact sequence of text tokens. Formally, we aim to maximize the conditional log-likelihood of the target text sequence $\mathbf{y} = (y_1, y_2, \cdots, y_T)$ given the prepended soft prompt $\mathbf{p} \in \mathbb{R}^d$. With a decoder-only LLM parameterized by $\Theta$, the optimization objective can be expressed as:

$$\max_{\mathbf{p}, ||\mathbf{p}||=B} \mathcal{L}(\mathbf{p}) = \max_{\mathbf{p}, ||\mathbf{p}||=B} \sum_{t=1}^{T} \log P(y_t|\mathbf{p}, y_{<t}; \Theta), \tag{6}$$

where $P(y_t|\mathbf{p}, y_{<t}; \Theta)$ denotes the probability of the token $y_t$ at time step $t$, conditioned on the soft prompt $\mathbf{p}$ and the previously generated tokens $y_{<t}$. Though more advanced algorithms could be applied, we find that the simple gradient ascent suffices to optimize Eq. 6, and the norm constraint is enforced by setting every iteration's result to a fixed norm $B$ (for simpler inverse mapping, discussed below). In addition, we empirically observe that tuning a single-token soft-prompt vector is sufficient to condition the selected LLM (Llama-3.1-8B, Dubey et al.) to output the exact sequence up to hundreds of tokens (within fewer than 1K optimization steps). After the optimization stage, we can then transform the soft-prompt embedding $\mathbf{p}$ to the target color distribution $\mathbf{h}^{tgt} \in \mathbb{R}^d$ thorough a simple deterministic mapping: $\mathbf{h}^{tgt} = f(\mathbf{p}) = \frac{\exp(\mathbf{p})}{\sum_i \exp(\mathbf{p}_i)}$, where the exponential function ensures the non-negativity and the normalization ensures all the entries of $\mathbf{h}^{tgt}$ sum to 1. As for the inverse mapping $f^{-1}$, we are essentially looking for some scaling factor $k \in \mathbb{R}$ such that $||\ln(k\mathbf{h}^{tgt})|| = ||\mathbf{p}||$. Given the fixed norm $B$, the value of $k$ can be efficiently calculated. Hence, we can always find a unique solution that constructs the one-to-one mapping between $\mathbf{p}$ and $\mathbf{h}^{tgt}$, where no additional information is required for decoding.

Figure 4: Qualitative results on color-constrained generation. "LoRA+CN+IP" refers to the stacked control from LoRA (Ruiz et al., 2023), ControlNet (Li et al., 2025), and IP-Adapter (Ye et al., 2023). HistKL quantifies the KL divergence to the target color distribution (the lower the better).

| Method | Primary Task | HistKL ↓ | CLIP ↑ | Aesthetics ↑ | Training-free? |
|---|---|---|---|---|---|
| Unconstrained | Text-to-Image Generation | 10.87 | 29.47 | 6.98 | - |
| LoRA+CN+IP | Color/Sturcture/Style Control | 3.16 | 22.53 | 5.39 | ✗ |
| PixelShuffler | Color Scheme Matching | 5.90 | 23.56 | 4.87 | ✗ |
| MPGD | Constrained Generation | 4.24 | 24.46 | 4.94 | ✓ |
| GPT-4o-Image | In-Context Generation | 8.96 | 25.72 | 6.62 | ✓ |
| InST | Style Transfer | 5.67 | 17.41 | 4.83 | ✗ |
| StyleID | Style Transfer | 11.32 | 24.76 | 6.18 | ✓ |
| StyleShot | Style Transfer | 1.90 | 26.39 | 5.83 | ✓ |
| InstantStyle++ | Style Transfer | 2.72 | 26.23 | 6.54 | ✓ |
| **Ours (direct OT)** | Img-to-Img Translation | **0.00** | 26.94 | 6.52 | ✓ |
| **Ours (guidance)** | Constrained Generation | 1.09 | **27.19** | **6.78** | ✓ |
| **Ours (guidance + post-hoc OT)** | Constrained Generation | **0.00** | 26.91 | 6.66 | ✓ |

Table 1: Quantitative results on color-constrained generation. The best results are shown in **bold**. The "direct OT" variant applies OT-based transformation to the unconstrained images; whereas the "guidance" variants apply OT-based guidance transformations during the sampling process.

## 5 EXPERIMENT

### 5.1 IMPLEMENTATION DETAILS

**Histogram Configuration.** For color-constrained generation with reference images, we use $d = 4096$ and single-option binning, where the color channels are quantized to match the histogram dimension (e.g., $16^3$ for RGB binning, $64^2$ for RG binning, etc.). For information embedding, we employ Llama-3.1-8B (Dubey et al., 2024) for soft prompt tuning, with $d = 4096$, fixed norm $B = 40.0$. We use the AdamW (Loshchilov & Hutter, 2019) optimizer with step-based learning rate decay. Empirically, the soft prompt for most text sequences of below 500 tokens converges within 500 optimization steps. For multi-option binning, we use $k = 16$ and $d = 4096$, resulting in 65536 color options. All cost matrices are pre-computed based on the L1 distance. We adopt the OT solvers offered by Python Optimal Transport (POT) [1].

**Image Generation.** To compare with relevant baselines fairly, we use SDXL (Podell et al., 2023) as the base model for image generation (on $1024^2$ resolution), with a DDIM (Song et al., 2021) noise scheduler and 50 sampling steps. Meanwhile, HIG generalizes to newer models trained with flow-based objectives (Lipman et al., 2022; Liu et al., 2023). The guidance time steps $\mathcal{T}$ are set by the usage. For histograms derived from reference images, 1-2 OT-based transformations are sufficient to offer descent distributional guidance. For the synthetic histograms derived from the soft-prompt embeddings, we generally need 3-4 OT-based transformations to get better generation quality. We direct the readers to the Appendix for more details on the generalization to flux.1[dev] (Labs, 2023) and the ablation studies on the optimal choice of $\mathcal{T}$.

---

[1] https://pythonot.github.io/

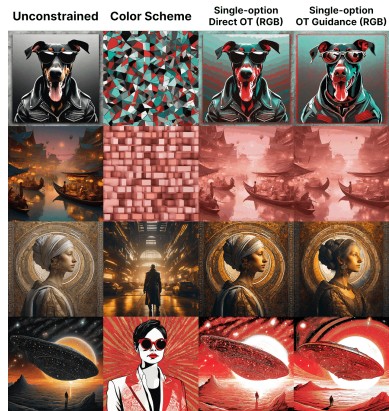

Figure 5: Qualitative comparison of color-constrained image generation using different HIG variants.

Table 2: Latency of different control baselines. By default, we use 50 sampling steps and FP16 precision on a single NVIDIA A100 GPU. * indicates sample-specific tuning.

| Method | Base Model | Latency (s) ↓ | Overhead (s) ↓ |
|---|---|---|---|
| Unconstrained | SDXL | 10.67 | – |
| HIG (w/o post-hoc OT) | SDXL | 12.87 | 2.20 |
| HIG (w/ post-hoc OT) | SDXL | 15.06 | 4.39 |
| DreamBooth LoRA* | SDXL | 13.01 | 2.34 |
| ControlNet++ (Depth) | SDXL | 25.47 | 14.80 |
| ControlNet++ (Softedge) | SDXL | 15.51 | 4.84 |
| ControlNet++ (OpenPose) | SDXL | 17.19 | 6.52 |
| InstantStyle++ | SDXL | 79.52 | 68.85 |
| InST* | SD1.4 | 4.88 | N/A |
| MPGD | SD1.4 | 10.83 | N/A |
| StyleID | SD1.4 | 23.89 | N/A |
| StyleShot | SD1.5 | 44.91 | N/A |
| PixelShuffler* | N/A | 38.42 | N/A |

## 5.2 EVALUATION ON COLOR-CONSTRAINED GENERATION

We compare the color-constrained generation capability of HIG with other relevant baselines, including GPT-4o-Image (OpenAI, 2025), stacked controls using DreamBooth LoRA (Ruiz et al., 2023), ControlNet (Zhang et al., 2023a), IP-Adapter (Ye et al., 2023), soft-shuffling guided color scheme matching (Zamzam, 2024), manifold preserving guidance He et al. (2024), and style transfer approaches (Zhang et al., 2023b; Chung et al., 2024; Gao et al., 2024; Wang et al., 2024). The curated benchmark consists of 300 textual prompts (sourced from Civitai [2]) and 300 reference images (half depicting certain content and half composed of abstract color schemes). Our evaluation includes both qualitative (Fig. 4) and quantitative (Tab. 1) aspects, measuring histogram alignment (HistKL), prompt compliance (CLIP-Score, Hessel et al.), and visual aesthetics (LAION-Aesthetics, Schuhmann et al.). Overall, the results show that HIG consistently outperforms other baselines in terms of histogram alignment, prompt compliance, and visual quality, regardless of whether post-hoc OT is applied. In addition, we observe that applying OT-based guidance transformations during the denoising process yields notable gains over the "direct OT" variant. As demonstrated in Fig. 5 (row 1), performing OT-based guidance during sampling helps alleviate the visual artifacts caused by the locally inconsistent pixels. Lastly, we evaluate the overheads incurred by different control baselines (Tab. 2). Overall, HIG incurs minimal computational overhead and is roughly comparable to common control methods like LoRAs and ControlNets.

## 5.3 EVALUATION ON INFORMATION EMBEDDING

The evaluation on histogram-based information embedding consists of two stages: in the first stage, we perform Prompt Tuning (Lester et al., 2021) to optimize for the soft-prompt embedding that can condition the LLM to output the exact text, where we observe the success rate of finding desired soft-prompts; in the second stage, we convert the soft-prompt embeddings to color histograms and perform constrained generation under such guidance, where the objective is to monitor the success information decoding rate based on the color histograms of the generated images.

To obtain complex text sequences to embed in images, we collect the Markdown code from popular GitHub repositories [3], providing a mixture of multilingual text, code, and URLs. We randomly crop the raw text to have a fixed number of tokens within $\{32, 64, 128, 256, 512\}$, resulting in 300 text sequences at each token length. For constrained generation, we adopt $\mathcal{T} = \{40, 30, 20, 10, 0\}$ (i.e., with post-hoc OT) to improve content fidelity and secure precise histogram alignment. By the qualitative results in Fig. 6, we observe that: 1) given the synthetic color histograms derived from soft prompt embeddings, performing direct OT on the unconstrained image tends to give coarse and noisy output under single-option binning (col2&4); 2) applying OT-based guidance during the denoising process significantly improves the generation quality while ensuring histogram alignment

---

[2]https://civitai.com/

[3]https://github.com/EvanLi/Github-Ranking

Figure 6: Qualitative results on information embedding via color histograms with different HIG variants and binning channels (RGB/RG).

| Text Tokens | Soft Prompt Tuning | | Binning Strategy | Decoding Exact Match ↑ |
|---|---|---|---|---|
| | Success ↑ | Time (s) ↓ | | |
| 32 | 100.0% | 4.87 | Single | 100.0% |
| | | | Multiple | 100.0% |
| 64 | 99.7% | 6.19 | Single | 99.7% |
| | | | Multiple | 99.7% |
| 128 | 99.7% | 11.16 | Single | 99.7% |
| | | | Multiple | 99.7% |
| 256 | 99.7% | 20.06 | Single | 97.3% |
| | | | Multiple | 97.7% |
| 512 | 98.3% | 300.08 | Single | 91.0% |
| | | | Multiple | 90.3% |

Figure 7: Quantitative results on information embedding. The "single" variant uses OT guidance, and the "multiple" variant adopts direct OT. Both variants compute histograms from RGB channels.

(col3&5); 3) direct OT with multi-option binning results in minor content deviations from the source image (col6), which is at the courtesy of relaxing the color-level constraint to bin-level.

The quantitative results are shown in Tab. 7. In general, the optimization of a text sequence up to 256 text tokens takes less than 20 seconds with a >99.7% success rate (evaluated on a single NVIDIA A100 GPU). For the decoding side, the exact match rate of both binning strategies slightly decreases as we enlarge the text sequence length. In the case of 512 tokens, the rate of decoding the exact target text for both binning strategies remains above 90%. Note that it is empirically possible to further improve the exact decoding rate by enlarging the image resolution or using multiple soft-prompts. Meanwhile, enlarging the embedded text length does not have a significant impact on the image quality. This is because the soft-prompt embedding is of fixed dimension and always converts to a random color distribution, whereas the exact decoding rate may saturate at certain text length due to the approximation capability with a finite number of pixels. Overall, the quantitative results on information embedding suggest that the proposed framework enables precise control over color histograms. We provide the robustness analysis in the Appendix.

## 5.4 GENERALIZATION TO LATENT HISTOGRAMS

To further examine the versatility of HIG, we extend our framework to discrete latent histograms. Specifically, we encode intermediate image predictions, manipulate the latent token maps based on a guidance image, and decode the perturbed tokens to obtain the transformed output. We apply this workflow across multiple image tokenizers, including VQ-GAN (Esser et al., 2021), TokenFlow (Qu et al., 2024), and TiTok (Yu et al., 2024). As illustrated in Fig. 8, the observed control effects are closely tied to the properties of the latent space. For tokenizers that capture low-level visual features (e.g., colors and textures), HIG produces effects akin to color scheme matching. In contrast, when operating in tokenizers that encode high-level semantics, manipulation of the latent histograms yields semantic-level guidance. Lastly, for 1D tokenizers, we observe strong control over spatial structures. Together, these findings demonstrate HIG's flexibility and generalization across diverse latent representations.

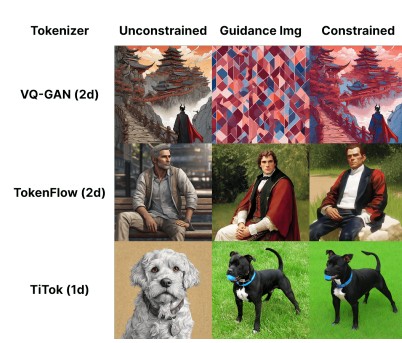

Figure 8: Varying control effects when manipulating latent token histograms from different image tokenizers.

## 6 RELATED WORK

**Text-to-Image Diffusion Models.** Following Rombach et al., recent approaches follow the LDM framework, with a trend of switching to the DiT-style architectures (Peebles & Xie, 2022) and scal-

ing up the models (Podell et al., 2023). Many recent works have incorporated new techniques for model training, such as flow matching (Lipman et al., 2022; Liu et al., 2023) and EDM (Karras et al., 2022; 2024), resulting in a series of competitive models (Esser et al., 2024; Labs, 2023; Chen et al., 2024; Xie et al., 2024; Gao et al., 2025). We note that our framework is widely applicable to any text-to-image model. As long as it requires multiple steps during sampling, we can always perturb the intermediate predictions and resume the generation in the same spirit.

**Controllable Generation.** There have been a variety of works that aim to introduce additional control signals. LoRA-based approaches utilize parameter-efficient fine-tuning (PEFT) to learn adaptation parameters that specialize the content or style from reference images (Hu et al., 2022; Ruiz et al., 2023; Liu et al., 2024; Shah et al., 2025). ControlNet-like approaches (Zhang et al., 2023a; Zhao et al., 2024; Li et al., 2025; Mou et al., 2024) introduce extra parameters to enforce fine-grained control over local regions based on dense spatial signals. Image colorization (Zamzam, 2024; Ke et al., 2023; Liang et al., 2024) and style transfer (Chung et al., 2024; Gao et al., 2024; Zhang et al., 2023b; Wang et al., 2024) approaches transfer the color scheme or artistic styles from reference images to other images. Some recent works take multimodal inputs and perform in-context image generation and editing (OpenAI, 2025; Labs et al., 2025; Wang et al., 2025).

**Training-free Guidance.** The OT-based guidance in HIG can be viewed as a form of training-free guidance (TFG). Most TFG approaches rely on heuristic-based gradients (Esser et al., 2021; Chung et al., 2022a; Bansal et al., 2023; Ye et al., 2024), whereas HIG operates in a possibly non-differentiable way by inserting a user-defined transformation into the loop. There are also derivative-free techniques that optimize non-differentiable targets by sampling multiple at each step and selecting ones with the highest rewards (Li et al., 2024; Huang et al., 2024; Dou & Song, 2024).

**Constrained Generation.** Our approach shares conceptual similarity with inverse problem methods (Chung et al., 2022b; 2023; He et al., 2024) and other constrained generation approaches (Fishman et al., 2023; Naderiparizi et al., 2024; Christopher et al., 2024), but with different constraints and task formulations. Overall, HIG adopts a common practice in constrained generation: projecting the unconstrained predictions to the feasible domain. We adapt this idea to the LDM (Rombach et al., 2022) framework and develop an OT-based formulation for enforcing histogram constraints.

**Information Embedding.** Prior works have focused on hiding data within some medium to avoid detection (also known as steganography). Most of them aim to bit-form information within generated images (Zhou et al., 2023b; Peng et al., 2023; Su et al., 2024; Zhou et al., 2023a). Unlike most existing methods that are restricted to dataset-specific scenarios, our approach is broadly applicable to any text-to-image model and any form of text by leveraging pre-trained LLMs and soft prompts.

## 7 DISCUSSION

While our histogram-constrained generation framework enables precise distributional control, several potential improvements are available: 1) For simplicity of implementation, we apply OT-based guidance transformations by randomly sampling pixels or tokens from each source bin and setting them to the target bin values. Better transport strategies are available, such as taking spatial or structural locality into account during sampling, which may reduce the visual artifacts after OT-based transformation; 2) While explicit transformation offers strong control guarantees, it may potentially introduce sharp transitions or inconsistencies. This motivates further exploration of softer designs, such as interpolating between pre- and post-perturbation representations.

## 8 CONCLUSION

We presented HIG, a novel framework for histogram-constrained image generation that enables precise, interpretable control in diffusion models by enforcing distributional constraints over pixels or latent tokens. HIG applies explicit OT-based guidance transformations during the denoising process, guiding the generation towards a user-specified histogram. Through extensive experiments, we demonstrate the versatility of HIG, including constrained generation with histogram guidance and high-capacity information embedding. Our results show that HIG achieves strong alignment with target distributions while preserving visual fidelity, offering a flexible control mechanism that complements existing approaches. We believe HIG provides a principled foundation for distributional control and opens new directions for hybrid and fine-grained controllable image synthesis.

## ETHICS STATEMENT

Our method shares the common advantages (e.g., creative content generation) and drawbacks (e.g., harmful or misleading outputs, copyright concerns) of other generative techniques. Beyond these, our information embedding technique enables encoding arbitrary text within visually indistinguishable images, making the hidden content resistant to detection or decoding without access to the exact decoder LLM. While this opens opportunities for secure data transmission, it also raises risks of misuse. A potential safeguard is to train classification models to identify images that are likely to contain embedded information, thus providing an additional layer of defense.

## REPRODUCIBILITY STATEMENT

The implementation details are discussed in Sec. 5.1. We will open-source our code on GitHub. Our anonymized repository is available at: `https://anonymous.4open.science/r/HIG`.

## USAGE OF LLMS

Large Language Models (LLMs) were used solely as auxiliary tools in preparing this work. Their involvement was limited to polishing the writing for clarity and style, as well as generating helper code snippets (e.g., for visualization or checking function usage). All core contributions (including developing methodologies, implementing the framework, conducting experiments, and analyzing results) were entirely the work of the authors.

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

# A  PSEUDOCODE FOR HISTOGRAM-CONSTRAINED IMAGE GENERATION

---

**Algorithm 1** Text-to-Image Generation with Explicit Guidance Transformation

---

1: **Input:** guidance transformation $\varphi$, guidance time steps $\mathcal{T}$, T2I model $\Theta$, text condition $\mathbf{c}$
2: $\mathbf{z}_T = $ sample-Gaussian-noise()
3: **for** time step $t = T$ **to** 1 **do**
4:    $\mathbf{z}_0^t = \frac{1}{\sqrt{\bar{\alpha}_t}}\mathbf{z}_t - \frac{\sqrt{1-\bar{\alpha}_t}}{\sqrt{\bar{\alpha}_t}}\epsilon_\theta(\mathbf{z}_t, \mathbf{c}, t; \Theta)$
5:    **if** $t \in \mathcal{T}$ **then**
6:      $\mathbf{z}_0^{t'} = \text{VAE-encode}(\varphi(\text{VAE-decode}(\mathbf{z}_0^t)))$
7:      $\mathbf{z}_{t-1} = \sqrt{\bar{\alpha}_{t-1}}\mathbf{z}_0^{t'} + \sqrt{1 - \bar{\alpha}_{t-1}}\epsilon_\theta(\mathbf{z}_t, \mathbf{c}, t; \Theta)$
8:    **else**
9:      $\mathbf{z}_{t-1} = \sqrt{\bar{\alpha}_{t-1}}\mathbf{z}_0^t + \sqrt{1 - \bar{\alpha}_{t-1}}\epsilon_\theta(\mathbf{z}_t, \mathbf{c}, t; \Theta)$
10:   **end if**
11: **end for**
12: $\mathcal{I}^{gen} = \text{VAE-decode}(\mathbf{z}_0)$
13: **Output:** generated image $\mathcal{I}^{gen}$

---

Listing 1: OT histogram matching (Python-style pseudocode)

```python
# Config:
# - Single-option binning with RGB inputs
# - Histogram of dimension 4096 (16^3, 4-bit quantization per channel)

def OT_histogram_matching(I_u8, h_tgt, num_bins=4096):

    h_src = calculate_histogram(I_u8)
    gamma = solve_OT_plan(h_src, h_tgt)

    left_4bits, right_4bits = split_bits(I_u8)
    bid = color_to_bin_id(left_4bits) # (r<<8)|(g<<4)|b in [0..4095]
    source_indices = indices_by_bin(bid) # {i: [indices...]}

    I_out = deepcopy(I_u8)
    for src_bin in range(num_bins):
        idx_pool = source_indices[src_bin]
        if is_empty(idx_pool):
            continue
        need = sum_nonzero(gamma[i])
        chosen = sample_without_replacement(idx_pool, need)
        start = 0
        for tgt_bin in nonzero_bins(gamma[src_bin]):
            cnt = min(gamma[src_bin][tgt_bin], len(chosen) - start)
            sl = chosen[start:start+cnt]
            color_tuple_4bits = bin_id_to_color(tgt_bin)
            I_out[sl] = (color_tuple_4bits << 4) + right_4bits[sl]
            start += cnt
    return I_out
```

## B  COMPATIBILITY WITH OTHER CONTROL MECHANISMS

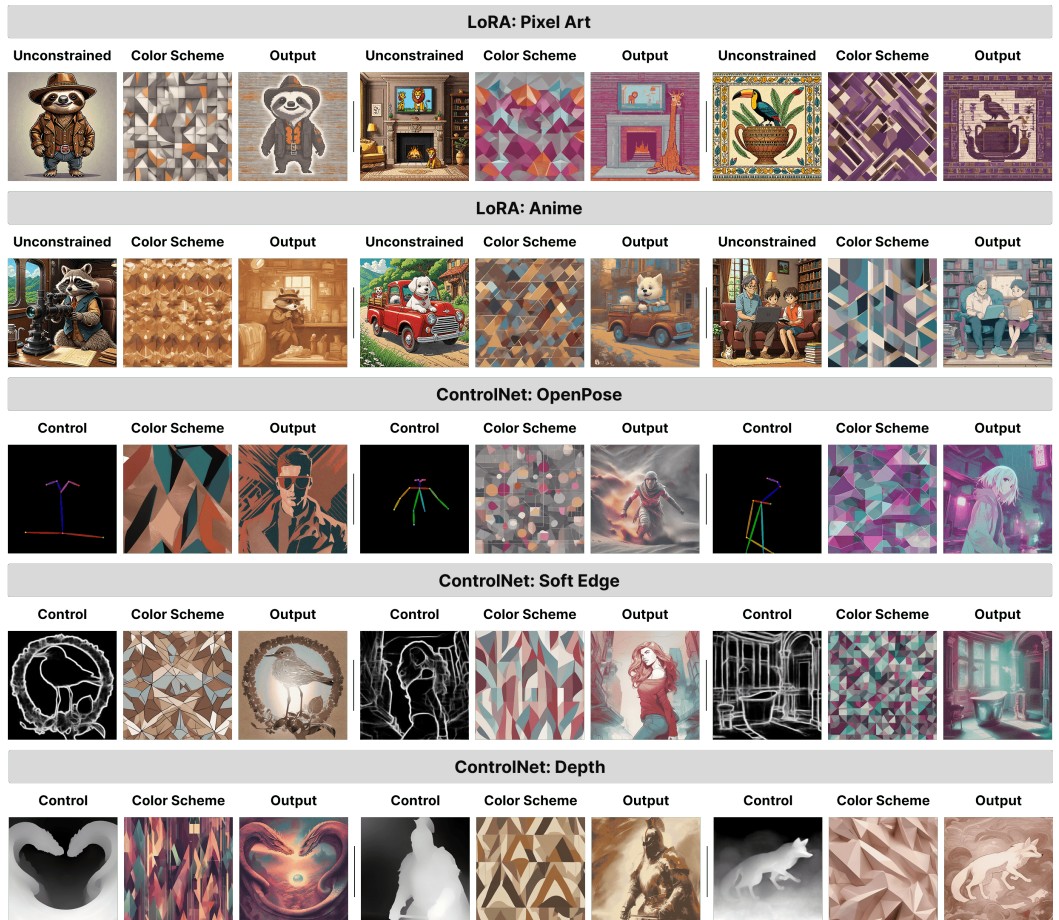

Figure 9: Qualitative results for combining HIG's distributional color control with DreamBooth LoRA Ruiz et al. (2023) and ControlNet++ Li et al. (2025). Better view with color.

## C  VISUALIZATION OF OT-BASED GUIDANCE TRANSFORMATION

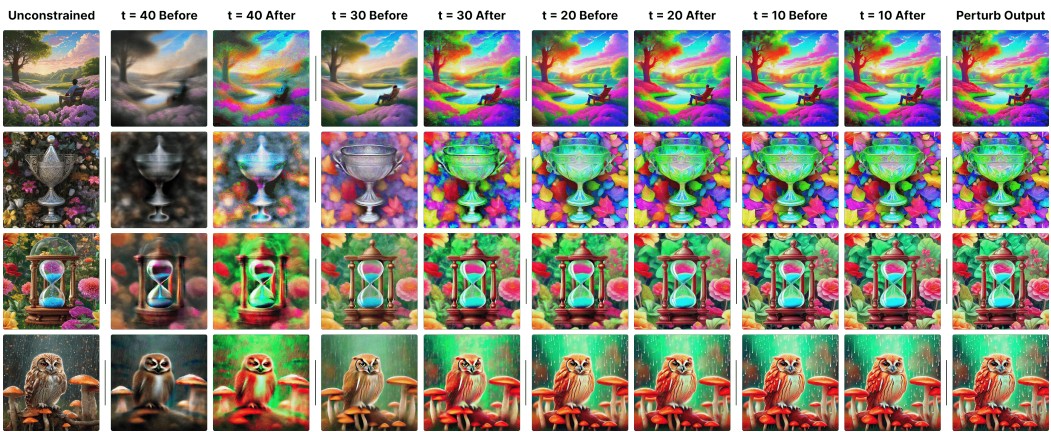

Figure 10: Visualizations of OT-based color histogram matching during the denoising process ($\mathcal{T} = \{40, 30, 20, 10\}$). For each example, we sample a random color histogram as $\mathbf{h}^{tgt}$. Row 1&2 use single-option binning on RGB channels; Row 3&4 use single-option binning on RG channels.

## D  CONTENT STABILITY AFTER MULTIPLE DECODING-ENCODING CYCLES

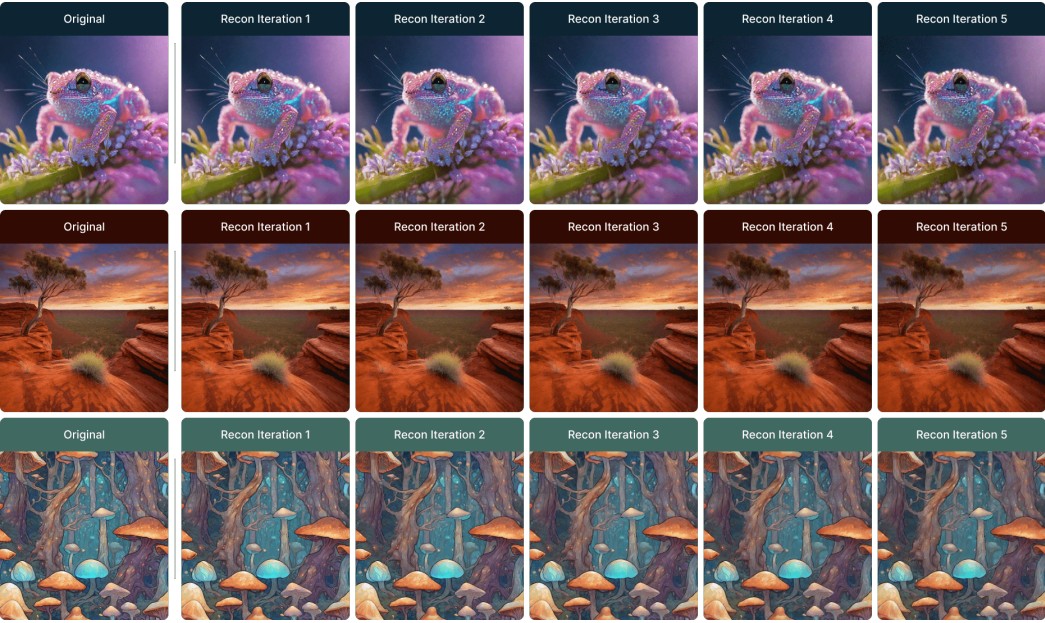

Figure 11: Content stability of SDXL VAE Podell et al. (2023) after multiple decoding-encoding cycles. Overall, the reconstructed images remain visually identical across cycles, demonstrating the feasibility of our decode-transform-encode diffusion guidance scheme.

# E  COLOR-CONSTRAINED GENERATION: FLOW-BASED MODEL

| Method | HistKL ↓ | CLIP ↑ | Aesthetics ↑ |
|---|---|---|---|
| Unconstrained | 9.65 | 28.76 | 7.16 |
| Ours (direct OT) | **0.00** | 26.01 | 6.92 |
| Ours (guidance w/o post-OT) | 1.34 | **26.76** | **7.04** |
| Ours (guidance w/ post-OT) | **0.00** | 26.13 | 6.98 |

Table 3: Porting HIG to **Flux.1-dev** (Labs, 2023), a flow-based generative model. The adaptation requires computing intermediate predictions under the flow objective. Results show that guidance-based HIG achieves improved aesthetics, while post-hoc OT enforces exact histogram matching (HistKL = 0). Overall, the observed trends are consistent with those on SDXL.

| Method | Latency (s) ↓ | Overhead (s) ↓ |
|---|---|---|
| Flux.1-dev baseline | 23.38 | - |
| HIG (w/o post-OT) | **26.59** | **3.21** |
| HIG (w/ post-OT) | 28.27 | 4.89 |
| DreamBooth LoRA | 28.21 | 4.83 |
| ControlNet (Canny) | 28.81 | 5.43 |

Table 4: Sampling latency and computational overhead on **Flux.1-dev** using 50 sampling steps and BF16 precision on a single NVIDIA A100 GPU. The results show that HIG introduces minimal overhead, comparable to DreamBooth LoRA and ControlNet baselines.

# F  COLOR-CONSTRAINED GENERATION: ABLATIONS ON GUIDANCE STEPS

As shown in Tab. 5, performing OT-based guidance at earlier diffusion steps yields better CLIP and aesthetic scores but weaker histogram alignment, while later guidance improves alignment at the cost of visual quality. Applying multiple guidance achieves near-perfect histogram matching without post-hoc OT, but further degrades image quality. This suggests that for color histograms, one or two well-placed OT-based guidance suffices to balance control precision and perceptual fidelity.

| Guidance Time Steps | Without Post-hoc OT | | | With Post-hoc OT | |
|---|---|---|---|---|---|
| | HistKL ↓ | CLIP ↑ | Aesthetics ↑ | CLIP ↑ | Aesthetics ↑ |
| $T = \{\}$ | 10.87 | 29.47 | 6.98 | 26.94 | 6.52 |
| $T = \{40\}$ | 5.37 | **28.94** | **6.99** | 26.41 | 6.44 |
| $T = \{30\}$ | 2.55 | 28.09 | 6.82 | 26.57 | 6.53 |
| $T = \{20\}$ | 1.09 | 27.19 | 6.78 | **26.91** | **6.66** |
| $T = \{10\}$ | 0.43 | 26.88 | 6.71 | 26.48 | 6.57 |
| $T = \{20, 40\}$ | 0.54 | 26.45 | 6.78 | 26.03 | 6.57 |
| $T = \{20, 30, 40\}$ | 0.43 | 25.93 | 6.60 | 25.71 | 6.43 |
| $T = \{10, 20, 30, 40\}$ | **0.39** | 25.24 | 6.51 | 25.10 | 6.36 |

Table 5: Ablation studies on guidance time steps for color-constrained generation. We compare the results for both with and without post-hoc OT adjustment. The HistKL for the case with post-hoc OT is guaranteed to be 0.0 and is thus omitted. The best results are shown in **bold**.

# G INFORMATION EMBEDDING: ABLATION ON OT FREQUENCIES

| OT Frequency | N=1 | N=3 | N=5 | N=7 | N=9 | Unconstrained |
|---|---|---|---|---|---|---|
| CLIP-Score ↑ | **27.93** | 26.57 | 26.89 | 26.61 | 26.39 | 29.47 |
| LAION-Aesthetics ↑ | 6.31 | 6.41 | **6.68** | 6.53 | 5.97 | 6.98 |

Table 6: Ablation on OT frequency for information embedding. Increasing the number of OT guidance ($N$) improves aesthetics up to $N = 5$, but further OT steps degrade text fidelity (CLIP-Score). All experiments apply a post-hoc OT step to guarantee exact histogram alignment (HistKL = 0).

# H INFORMATION EMBEDDING: ROBUSTNESS ANALYSIS

As shown in Fig. 12, we evaluate the robustness of our information embedding technique from the following perspectives: random scaling, JPEG compression, and corrupted histograms. When scaling the images randomly, we measure the exact reconstruction rate under a factor uniformly sampled from $[0.5, 2.0]$. In general, the impact is not significant when embedding shorter text, for example, the success rate is around 90% for sequences up to 128 tokens, and the performance drop becomes larger for longer sequences. In comparison, the generated images are more sensitive to JPEG compression, where the success rate drops below 90% when the text exceeds 32 tokens. We also investigate the robustness under corrupted histograms, including wrongly binned values and noisy soft prompts. Overall, for less than 5% of wrongly-binned values or Gaussian noise still secures success decoding rates above 90% at the sequence length of 128 tokens.

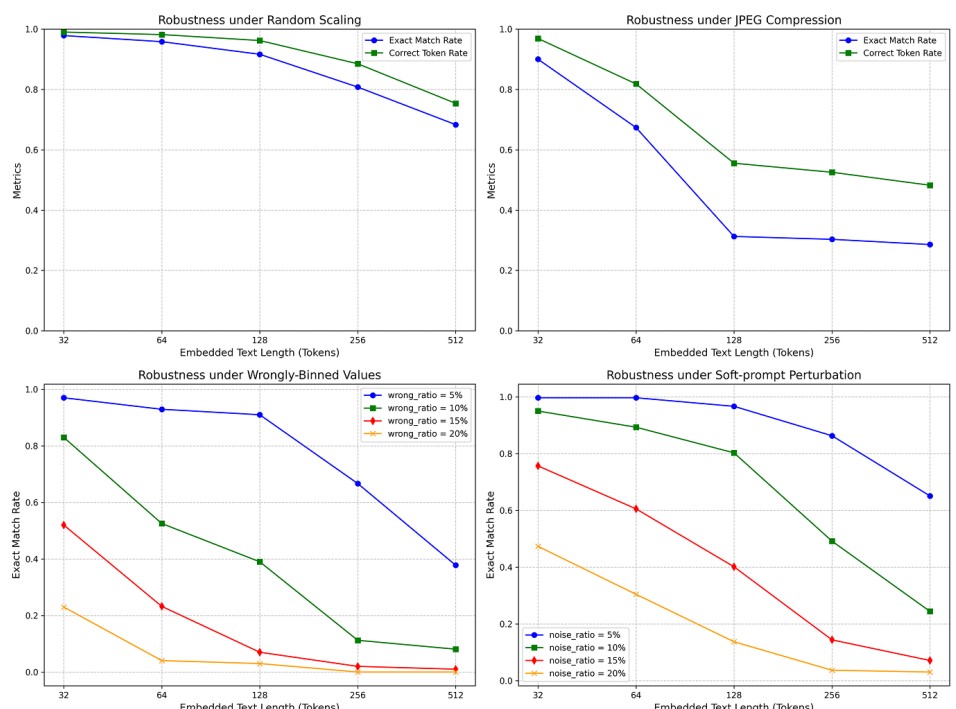

Figure 12: Robustness evaluation of our information embedding technique.

# I   INFORMATION EMBEDDING: SENSITIVITY TO INPUT RESOLUTION

| Embedded Text Tokens | 32 | 64 | 128 | 256 | 512 |
|:---:|:---:|:---:|:---:|:---:|:---:|
| $512^2$ | 100.0% | 99.3% | 98.0% | 94.7% | 87.7% |
| $1024^2$ | 100.0% | 99.7% | 99.7% | 97.3% | 91.0% |
| $2048^2$ | 100.0% | 99.7% | 99.7% | 98.0% | 94.3% |

Table 7: Decoding accuracy (%) for different input resolutions when embedding varying numbers of text tokens. Larger resolutions ($512^2$, $1024^2$, $2048^2$) provide finer histogram granularity and improve decoding rates, especially as the number of embedded tokens increases.

# J   INFORMATION EMBEDDING: HIGHLY COMPLEX INPUTS

Apart from the evaluation on Markdown code and normal images, we further evaluate its robustness under highly complex text and image signals. To do so, we generate images with intricate structures using the prompt "an artwork with intricate details, vibrant colors, high resolution, 8k" and construct strings with non-standard characters of length 128 (i.e., generally around 200 tokens) by randomly sampling from the UTF-8 encoding space. As shown in Fig. 13, our approach can still generate information-embedded images with high fidelity. Quantitatively, the rate of decoding exact text over 300 image-text pairs is slightly beyond 80% in both cases.

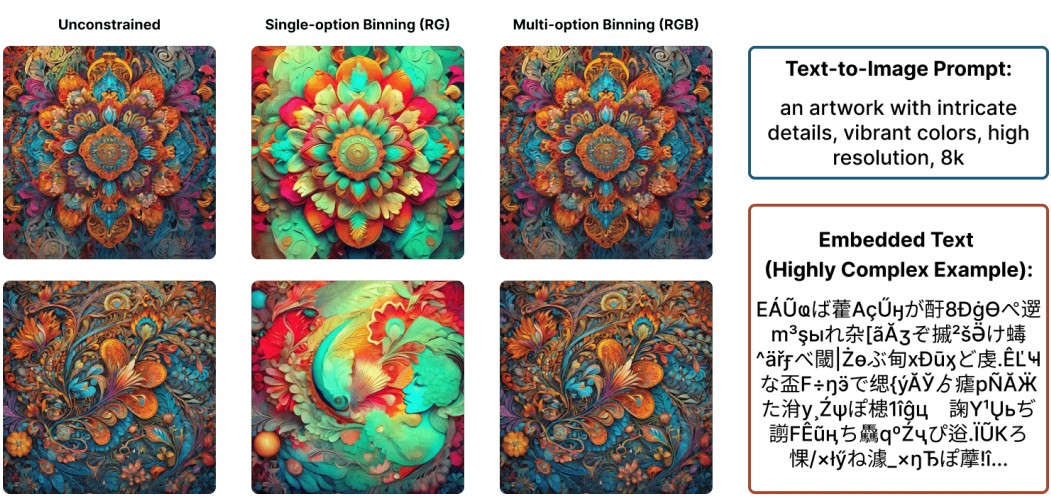

Figure 13: Robustness under highly complex embedded text and image content.

## K   LATENT HISTOGRAM: COLOR SCHEME MATCHING

| Method | Histogram | HistKL $\downarrow$ | CLIP $\uparrow$ | Aesthetics $\uparrow$ |
|---|---|---|---|---|
| Ours (guidance w/o post-OT) | latent token | 2.53 | 25.90 | 6.24 |
| Ours (guidance w/o post-OT) | pixel color | 1.09 | **27.19** | **6.78** |
| Ours (guidance w/ post-OT) | pixel color | **0.00** | 26.91 | 6.66 |

Table 8: Comparison between latent-token and pixel-space histogram matching. Latent histograms capture compressed representations but yield weaker control precision. Pixel histograms provide stronger alignment, with post-OT ensuring exact matching (HistKL = 0). Note that post-OT does not apply to latent histograms due to cycle-consistency issues.

## L   LATENT HISTOGRAM: SEMANTIC CONTROL

We explore guiding a simple unconstrained image (e.g., an empty background) toward the content of a semantically rich guidance image via a latent histogram constraint. Specifically, we first generate $N = 300$ random images with the prompt "an empty background", then apply constrained generation using latent histogram guidance from semantically rich images and a semantically neutral prompt "a picture". We evaluate the semantic control based on CLIP scores, including the alignment between both (image, ground-truth prompt) and (image, ground-truth image). Overall, latent histogram-constrained images show a clear improvement over the empty background baseline and achieve notable semantic alignment with the guidance image.

| Method | CLIP-Score (img, prompt$_{gt}$) $\uparrow$ | CLIP-Score (img, img$_{gt}$) $\uparrow$ |
|---|---|---|
| Empty Background | 9.51 | 56.14 |
| Latent-constrained Generation | 24.19 | 71.62 |
| Guidance Image (different seed) | 29.34 | 85.18 |
| Guidance Image (ground truth) | 29.47 | 100.00 |

Table 9: Evaluation of semantic control using latent histogram guidance.

## M    FAILURE CASE ANALYSIS: EFFECT OF POST-HOC OT

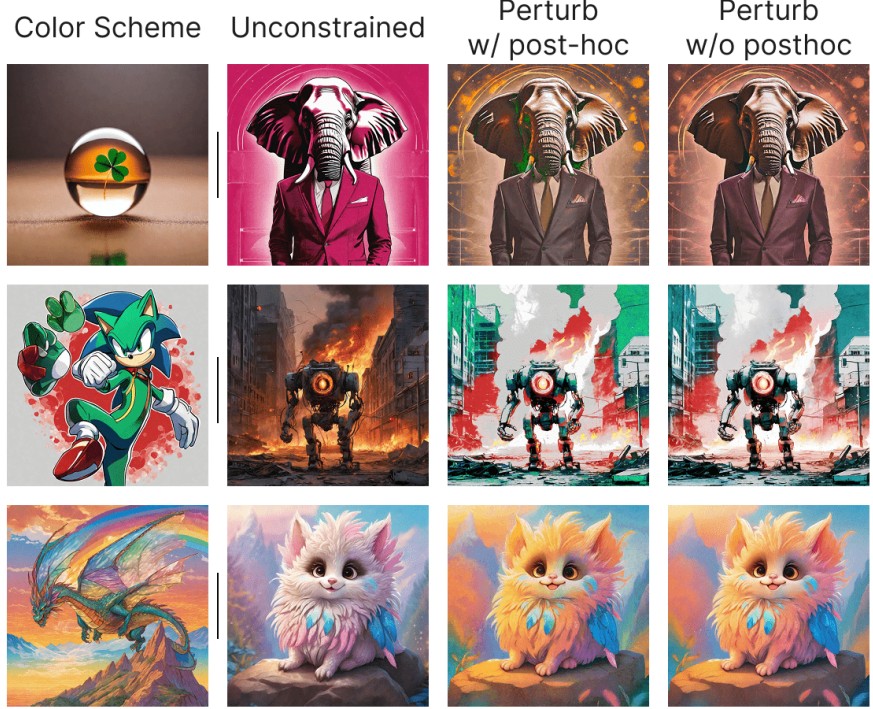

Figure 14: A post-hoc OT step can enforce exact compliance with $\mathbf{h}^{tgt}$, but may introduce visual artifacts from rigid color reassignment. While such strict control is essential for tasks like information embedding, it can be safely omitted in more flexible settings such as color scheme matching.

## N    EXTENDED USAGE: LIGHTING CONTROL WITH COLOR HISTOGRAMS

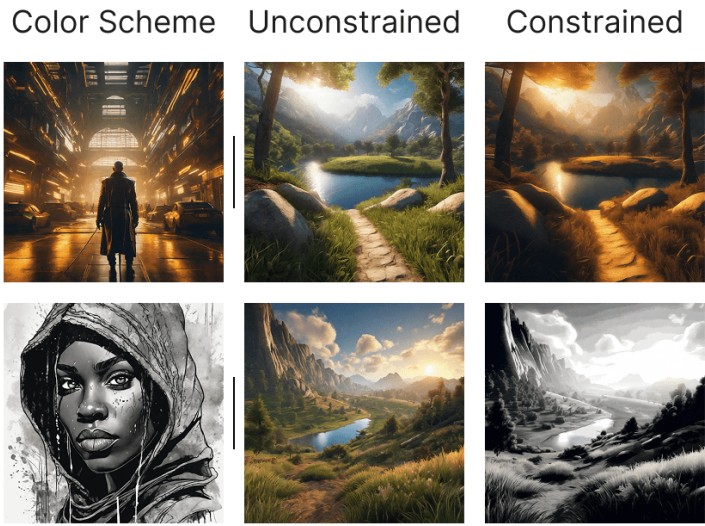

Figure 15: The color histogram constraint also enables lighting control on photo-realistic images.

# O ADDITIONAL QUALITATIVE RESULTS

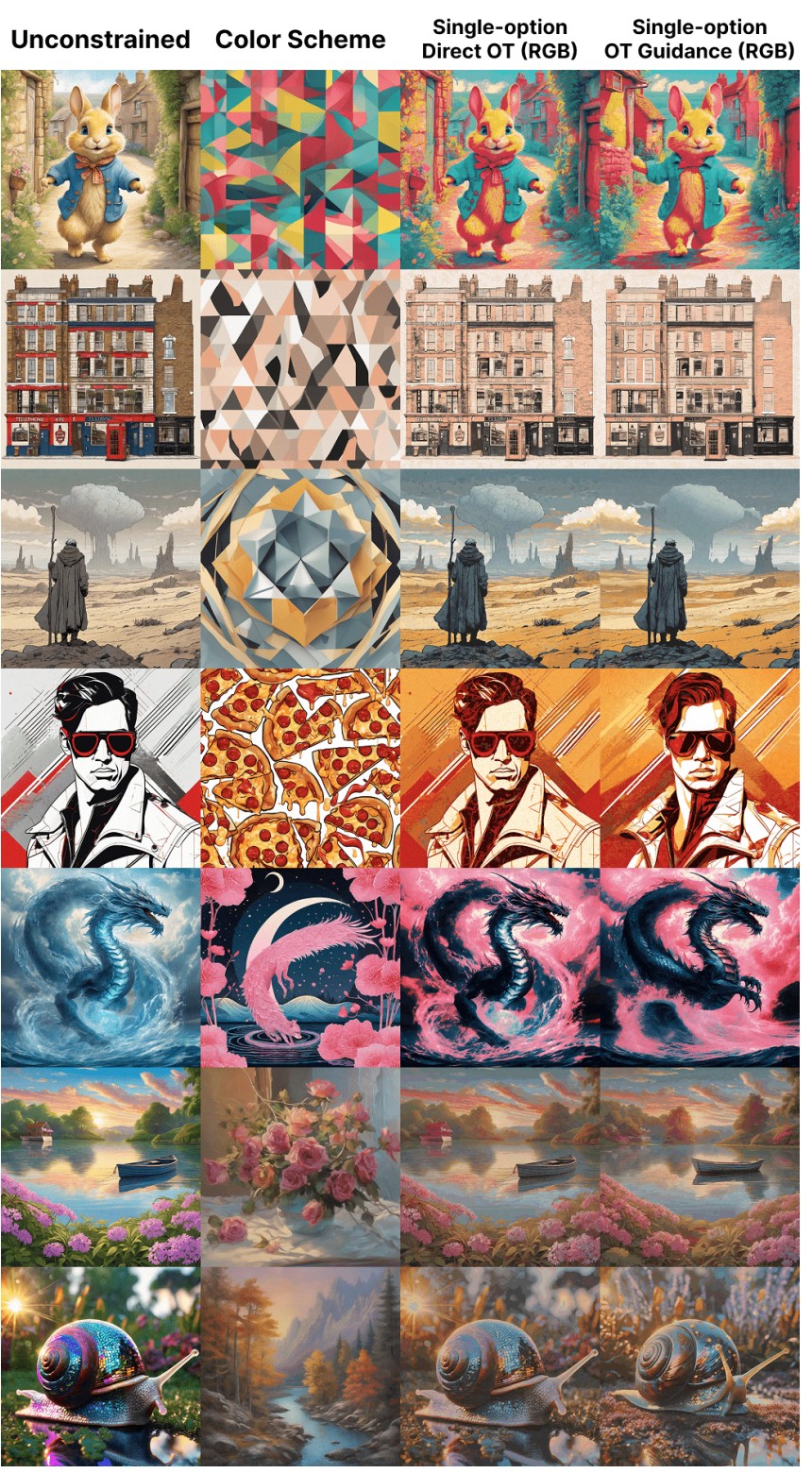

Figure 16: More qualitative results for color-constrained image generation (with post-hoc OT).

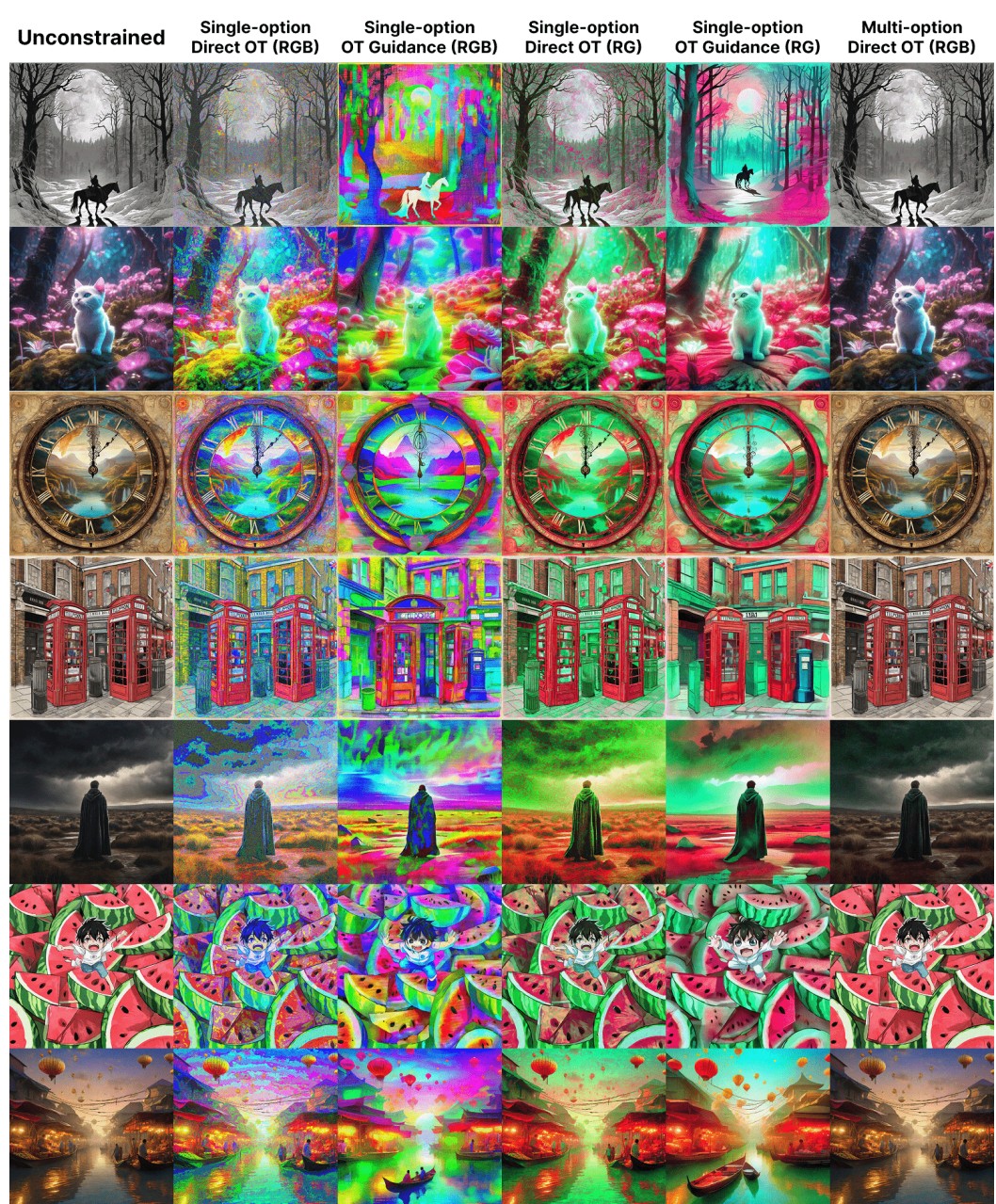

Figure 17: More qualitative results for information embedding via color histograms.

