# OpenReview forum: "Histogram-constrained Image Generation"
_ICLR.cc/2026/Conference — Submitted to ICLR 2026_

### Official Review · Reviewer_jNUJ · 2025-10-29

**Soundness:** 2
**Presentation:** 2
**Contribution:** 1
**Rating:** 4
**Confidence:** 4

**Summary:**

This paper proposes Histogram-Constrained Image Generation (HIG), a diffusion-based method that enforces user-specified histogram constraints (e.g., color or latent token distributions) during the image generation process. The constraint is modeled as an Optimal Transport problem and applied as explicit perturbations in the sampling process. The authors present this as a new form of “distributional control” between global semantic and local structural conditioning. Experiments show that HIG can produce images whose color histograms exactly match the given targets and can embed information via histogram encoding.

**Strengths:**

1. The proposed method is training-free and compatible with existing control methods, such as ControlNet or LoRA.

2. The method can enforce exact alignment between the generated images and the target histograms, as shown by the zero HistKL values in Table 1, which confirms the precision of the proposed constraint mechanism.

3. The paper is clearly written and easy to follow.

**Weaknesses:**

1. **Unclear application distinction and benefits**: While the paper claims that HIG ensures distributional consistency and enables several applications (line 89), two of these applications substantially overlap with existing tasks.

* Color-constrained generation appears conceptually similar to classical style transfer. The paper could better articulate what fundamental distinction HIG introduces, or why strict histogram alignment is particularly advantageous for this task. In terms of performance, in several examples (e.g., Fig. 5, first row, third image), histogram alignment introduces visible artifacts, while in others (e.g., Fig. 5, third row, last image), content consistency is compromised. These issues undermine the claimed controllability and visual fidelity.

* The information embedding task also largely overlaps with diffusion-based steganography methods such as DiffSteg [4] and HiDiffusion [5]. However, the manuscript reports results in isolation, without any quantitative or qualitative comparison to such baselines. It is unclear what unique benefits that histogram-based embedding provides compared to prior diffusion-based steganography approaches.

2. **Novelty concerns**: The proposed use of histogram matching for style transfer is not new — similar ideas have already been explored, such as [1]. In addition, employing Optimal Transport (OT) for color transfer and histogram alignment has been extensively studied in prior works [2][3]. This work mainly adapts these established concepts into the diffusion sampling loop, rather than introducing a novel formulation or demonstrating clear advantages over existing OT-based histogram matching approaches.

3. **Concerns about generalization and robustness**: The main experiments rely on SDXL, an outdated backbone. Although the authors claim HIG is compatible with newer DiT-based models, Appendix E only shows HIG's color transfer results on FLUX.1[dev], without comparing with state-of-the-art DiT-based style transfer methods or visualization. In addition, integrating HIG with DreamBooth produces noticeable identity inconsistence in the “LoRA anime” case (Appendix B, Fig. 9), but the paper did not analyze the cause of this degradation.



**References**

[1] Zhang, Y. et al., Exact Feature Distribution Matching for Arbitrary Style Transfer and Domain Generalization, CVPR 2022.

[2] Lim, F. et al., Order Constraints in Optimal Transport, ICML 2022.

[3] Larchenko, M. et al., Color Transfer with Modulated Flows, AAAI 2025.

[4] Zhang, H. et al., DiffSteg: Diffusion Model for Image Steganography, ICCV 2023.

[5] Wang, J. et al., HiDiffusion: Hiding Information in Diffusion Models for Steganography and Watermarking, CVPR 2024.

**Questions:**

See my comments in Weaknesses.

---

> ### Author Response · Authors · 2025-11-25
>
> Dear Reviewer jNUJ,
>
> We thank you for the time and effort in engaging with our work.
>
> ---
> > ### **Q: Application distinction & relations to style transfer**
>
> We wish to clarify that **HIG is not intended as a drop-in replacement for classic tasks like style transfer**. Instead, style transfer serves as a relevant baseline when evaluating HIG for color-constrained generation. The intended takeaway from Tab.1 is that HIG offers stronger color-constrained generation capabilities than existing control methods, including style transfer. **HIG provides capabilities not available in prior methods; while we demonstrate its versatility through several applications, the core contribution lies in the control paradigm itself.**
>
> Because HIG applies explicit guidance transformations during inference, it provides two distinct capabilities:
> (1) **Exact value control**: HIG can enforce predetermined color sets or distributions with pixel-level fidelity, offering substantially stronger controllability than related baselines (as demonstrated in the color-constrained generation experiments); (2) **Strict distributional conformity**: The ability to match target histograms enables downstream applications where fidelity to prescribed statistics is essential, such as the information embedding demonstrations in the paper. Moreover, HIG’s control paradigm is also **flexible**: the same mechanism can be adapted to region-specific constraints (via masks) or to alternative histogram families with minimal modification. Overall, we wish to underscore that **HIG introduces a complementary and uniquely precise form of control for diffusion-based generative models**.
>
> ---
> > ### **Q: Comments on qualitative examples**
>
> Regarding the examples in Fig. 5, we would like to clarify two points. First, the artifacts observed in the first row, third column, are expected: they arise from the *Direct OT* variant, which deliberately enforces rigid pixel-level color assignment only after generation. **This baseline is included precisely to illustrate that such post-hoc transformation may sometimes introduce noticeable artifacts**, whereas applying OT-based transformations during the denoising process (fourth column, our final approach) substantially mitigates these artifacts. Second, **for the perceived identity shift in the third row, the textual prompt specifies only vague identity attributes (i.e., “a woman”)**. Under such generic prompts, noticeable changes in facial characteristics can arise from simple random seed variations. Thus, the observed differences fall within the expected behavior of the underlying diffusion model rather than a failure of HIG’s controllability.
>
> Additionally, similar situations also apply to the examples in Appendix B, Fig. 9, where the textual prompts specify only vague identity attributes and are not expected to be consistent.
>
> ---
> > ### **Q: Relations to steganography works**
>
> First, we note that **the cited works “DiffSteg [4]” and “HiDiffusion [5]” do not appear to exist**. Among existing literature, the closest match is DiffStega [c1], which adopts a fundamentally different steganography formulation: it hides an image within another image using text as keys. In contrast, our setting aims to hide free-form textual messages inside generated images.
>
> While we discussed steganography methods with similar task formulations in the submission ([c2-c4]), a direct comparison is unfortunately infeasible. **These approaches are typically designed for dataset-specific settings (e.g., FFHQ, AFHQ) and rely on architectures or priors that do not generalize to open-domain generative models.** To the best of our knowledge, our method is the first to embed arbitrary text into open-domain images generated by text-to-image models.
>
> ---
> > ### **Q: Other related works**
>
> We summarize the key differences as follows: [1] operates on histograms constructed from continuously valued hidden states, which are implicit and require sophisticated bin-splitting strategies; whereas HIG's color-constrained setting focuses on explicit, concrete color histograms in pixel space. [2] develops an improved family of OT solvers with order constraints. Its color-transfer application relies on human-crafted priors and requires the "preferred solution" feedback from human users. [3] learns color distributions from implicit feature representations, which do not provide exact control over either the color values or the distributional properties, in contrast to HIG’s explicit and precise histogram enforcement.
>
> ---
>
> [c1] "DiffStega: Towards Universal Training-Free Coverless Image Steganography with Diffusion Models," Yang et al., 2024.
>
> [c2] "Secret-to-Image Reversible Transformation for Generative Steganography," Zhou et al., 2022.
>
> [c3] "StegaDDPM: Generative Image Steganography based on Denoising Diffusion Probabilistic Model," Peng et al., 2023.
>
> [c4] "StegaStyleGAN: Towards Generic and Practical Generative Image Steganography," Su et al., 2024.

---

### Official Review · Reviewer_7m5F · 2025-11-01

**Soundness:** 2
**Presentation:** 3
**Contribution:** 2
**Rating:** 4
**Confidence:** 3

**Summary:**

The authors proposed a new conditional image generation method that adds optimal transport steps to adjust color distribution during diffusion model inference. The target color histogram can be either extracted from a real reference image or implicitly obtained from optimizing the so-called information embedding of an LLM. The authors showed that the proposed method can generate quality images with higher semantic and aesthetic scores, and advertised the efficient training-free nature of the method.

**Strengths:**

1. The paper is really well-presented. The generated images in the paper are visually appealing. Diagrams are clear. Most of the paper is easy to follow, except for the information embedding part. See my question below.
2. The proposed method outperforms the baselines both visually and quantitatively. Being efficient is also important for any practical usage of diffusion models.

**Weaknesses:**

1. My main concern for this paper is that, if 1-2 optimal transports are already sufficient for generating the target color distribution, this task might not be difficult enough. Looking at the reference images presented in the paper, they are almost all synthetic images with highly concentrated color histograms, meaning there are only a few colors to spread. How difficult is this transferring task? An important baseline should be directly applying optimal transport to the output without interfering with the diffusion model. The resulting images might have some artifacts, but one can simply run more denoising steps to fix them, which essentially falls into the 1 optimal transport step case of the method.
2. The perhaps more interesting case with latent histograms is less discussed. Figure 8 shows some harder reference images, but the results are less satisfying. For the TiTok model, the generated image did not preserve the unconstrained generation but became very similar to the reference image. This is against the authors' claim that "the transformation remains close to the original diffusion trajectory".

**Questions:**

1. Using the normalized information embedding as an implicit histogram feels weird to me. They are fundamentally different objects. Could the authors elaborate on the logic behind this? Did the authors specifically design the multi-option optimal transport to make the color histogram the same dimension as the token embeddings?

---

> ### Author Response · Authors · 2025-11-25
>
> Dear Reviewer 7m5F,
>
> We thank you for the time and effort in engaging with our work.
>
> ---
> > ### **Q: Variant comparison and application scenario**
>
> We thank the reviewer for suggesting potential variants. In fact, our paper already includes this comparison (termed “Direct OT”). As shown in Fig. 5 (col. 3), **directly applying OT to the generated image often introduces noticeable local artifacts; performing OT-based transformations during denoising (as done in HIG) helps alleviate this issue**. Table 1 further compares multiple HIG variants with relevant baselines and shows that our approach offers superior control over color distributions.
>
> We agree that **HIG is particularly effective for non-photorealistic domains** (e.g., illustrations, anime), where enforcing exact color sets or distributions is desirable. Meanwhile, we also showcase its performance on photorealistic images for lighting control in Appendix N. Additionally, the framework itself remains **flexible**: the same mechanism can incorporate region-specific constraints (via masks) or alternative histogram families with minimal changes. Overall, **HIG provides a complementary and uniquely precise form of control that is not achieved by direct OT or relevant control baselines**.
>
> ---
> > ### **Exploration of latent histograms**
>
> Our intent in introducing latent histogram guidance was primarily to provide a **proof-of-concept** demonstrating that the HIG paradigm naturally generalizes to discrete latent token spaces, and the effect of control depends on the type of information encoded in the latent representation (e.g., colors, semantics, etc.). Unlike pixel histograms, performing explicit transformations on latent tokens involves far more confounding factors. In particular, **different latent spaces exhibit varying levels of robustness, leading to potentially heterogeneous control behaviors**. Empirically, applying OT-based transformations in more robust latent spaces tends to result in mild deviations from the original trajectory, while less robust spaces may yield noticeably coarser or even collapsed outputs.
>
> Our experiments in Fig. 8 and Appendix L aim to convey this core message: **performing HIG in latent space is feasible, and its effects are representation-dependent**. A full exploration of control outcomes under different latent spaces would require a dedicated study, and we view this as a promising direction for future work.
>
> ---
> > ### **Logic behind information embedding**
>
> To clarify: **(1) The information embedding application primarily aims to demonstrate HIG’s control precision; (2) The color histogram dimension is indeed chosen intentionally to match the hidden dimension of Llama-3.1-8B (d=4096)**.
>
> The detailed workflow is as follows. We first condense the textual message into a compact 4096d soft-prompt vector. To embed this information into an image, we convert the soft prompt into a normalized probability distribution and pair each value with a corresponding bin in a 4096d color set (the matching dimensionality is intentional). This target color distribution is then enforced through HIG during generation. At decoding time, we extract the color histogram from the generated image, reconstruct the normalized vector, and feed it back as a soft prompt to the LLM to recover the original message. This demonstrates that HIG can enforce precise, distribution-level conformity, enabling applications where exact compliance is required, even though the histogram and embedding spaces are fundamentally different objects.

---

### Official Review · Reviewer_ZzyM · 2025-11-01

**Soundness:** 3
**Presentation:** 3
**Contribution:** 2
**Rating:** 6
**Confidence:** 4

**Summary:**

This paper introduces **Histogram-Constrained Image Generation (HIG)**, a training-free control framework for diffusion models that enforces user-specified distributional constraints (e.g., color histograms, latent token distributions) during sampling. HIG fills the "middle ground" of control granularity, between high-level text prompts and low-level dense signals (e.g., ControlNet edge maps), by modeling distributional alignment as an Optimal Transport (OT) problem. It applies minimal-cost OT-based transformations to intermediate diffusion outputs (via a decode-transform-encode cycle) to guide the generation trajectory toward the target histogram.

**Strengths:**

1. Optimal Transport Ensures Precision and Minimal Distortion
   1. OT’s minimal-cost property preserves visual quality.
   2. Multi-option binning mitigates content distortion.
   3. Post-hoc OT guarantees exact alignment.
2. Training-Free Design Enables Low Overhead and Compatibility
   1. Negligible inference overhead.
   2. Compatibility with existing controls.
   3. Generalization to flow-based models.
3. Diverse Applications Demonstrate Versatility
   1. Color-constrained generation.
   2. High-capacity information embedding.
   3. Latent histogram control.
4. Reproducibility and Transparency
   1. Detailed implementation.
   2. Failure case analysis.

**Weaknesses:**

1. Incomplete Analysis of OT and Binning Design Choices
   1. Bin count and channel choice lack sensitivity analysis. d=4096 is used for all experiments, but no tests on smaller or larger d or single-channel histograms (e.g., grayscale) are reported. Appendix K compares latent vs. pixel histograms but not bin count’s effect on control precision.
   2. Multi-option binning k-value selection is arbitrary. $k=16$ is used for multi-option binning (Section 5.1) but no ablations on $k\in{8,32}$ are provided. It is unclear if larger k improves fidelity or if smaller k reduces computation.
2. Mathematical and Notation Ambiguities
   1. OT cost matrix construction is underspecified. Section 3.2 states cost is based on "L1 distance between color tuples/latent embeddings" but does not clarify if latent embeddings are pre-trained (e.g., CLIP) or tokenizer-specific (e.g., VQ-GAN codebook). Appendix B’s pseudocode uses RGB L1 but not latent costs, creating ambiguity for latent histogram implementation.
   2. Soft-prompt to histogram mapping lacks rigor. Equation 6 (soft-prompt optimization) uses a fixed norm $B=40.0$, but no justification for this value is provided. The inverse mapping (Section 4.2) mentions "scaling factor k" but does not derive k’s uniqueness mathematically, leaving uncertainty about decoding reliability.
   3. Intermediate step selection (T) is heuristic. Table 5 shows $T={40}$ (early step) improves CLIP (27.19) while $T={10}$ (late step) improves HistKL (0.54), but no method for optimal T selection is proposed. Users must manually tune T, reducing practicality.
3. Limited Evaluation of Content Preservation and Generalization
   1. Content distortion in high-semantic latent spaces. Section 5.4 (TiTok) shows semantic control but no metrics for content preservation (e.g., CLIP alignment with original prompt vs. guidance image). It is unclear if latent OT distorts intended content while enforcing token histograms.
   2. Lack of user study for aesthetics. Aesthetics scores (LAION-Aesthetics) are used, but no user evaluations of perceptual quality (e.g., preference between HIG and StyleShot) are conducted. Quantitative metrics may not capture subjective judgments of "naturalness" after OT.

**Questions:**

1. **How do OT solver choice (simplex vs. Sinkhorn) and bin parameters (d, k) impact speed, alignment, and fidelity, and what guidance can be provided for tuning them?** The paper uses a vanilla simplex solver and fixed d=4096/k=16. Could you add a table comparing Sinkhorn (with entropic regularization) vs. simplex on SDXL, reporting HistKL, latency, and Aesthetics for $d\in{512,2048,4096}$ and $k\in{8,16,32}$? Additionally, could you provide a heuristic (e.g., "choose d=2048 for balanced speed/alignment") for users with different hardware constraints?
2. **How does HIG preserve intended content when applying OT to high-semantic latent spaces (e.g., TiTok), and can you quantify this with content-specific metrics?** It is a well-known problem of all granularity control methods that enforcing constraints may distort intended content. Section 5.4 shows semantic control via latent histograms but no content preservation measures.Could you add experiments where you enforce a latent histogram from a "cat" image onto a "dog" prompt, reporting CLIP scores for "dog" (content) and "cat histogram" (constraint)? This would clarify if HIG distorts intended content. Also, could you test if adding a content loss (e.g., CLIP between original and OT-transformed latents) mitigates distortion?
3. **Can you formalize the soft-prompt to histogram mapping (and inverse) and validate its uniqueness across diverse text sequences?** Section 4.2 uses a fixed norm B=40.0 and exponential mapping but no mathematical proof of uniqueness. Additionally, could you test decoding uniqueness by mapping two different soft prompts to histograms, generating images, and verifying that decoded prompts match the original (not swapped)?
4. **Can you conduct a user study to evaluate perceptual quality and naturalness of HIG-generated images compared to baselines?** While LAION-Aesthetics scores are reported, subjective judgments of "naturalness" may not align with quantitative metrics. Could you run a user study where participants rate images from HIG vs. StyleShot and other baselines on a Likert scale for naturalness and preference? This would provide stronger evidence of HIG’s perceptual quality.

Overall, the paper presents a technically sound framework for histogram-constrained image generation via Optimal Transport. I tend to accept the paper, but it would benefit from addressing incomplete analyses (OT solver choice, bin parameters), mathematical ambiguities (cost matrix, soft-prompt mapping), and content preservation evaluations (latent OT distortion). Addressing these questions would strengthen the contribution and practical guidance for users.

---

> ### Author Response · Authors · 2025-11-25
>
> Dear Reviewer ZzyM,
>
> We thank you for the time and effort in engaging with our work.
>
> ---
> > ### **Q: Additional ablations and comparisons**
>
> | Image resolution (HxW) | 1024×1024 | 2048×2048 | 4096×4096 |
> |-|:-:|:-:|:-:|
> | **Single-round OT solving latency (s) ↓** | 0.18 | **0.17** | 0.21 |
>
> With fixed histogram dimension (d = 4096), **increasing image resolution does not introduce notable overhead in OT solving**.
>
> | Histogram dimension (d) | d = 512 (3-bit) | d = 4096 (4-bit) | d = 32768 (5-bit) |
> |-|:-:|:-:|:-:|
> | **CLIP-Score ↑** | 26.74 | 27.19 | **27.26** |
> | **LAION-Aesthetics ↑** | 6.59 | 6.78 | **6.81** |
> | **Single-round OT solving latency (s) ↓** | **0.01** | 0.18 | 10.47 |
>
> Under fixed resolution (1024x1024) and larger histogram dimension (d), the OT computation becomes more expensive; however, we also found that **enlarging d beyond the default config offers only marginal visual benefits**. FYI, our default choice d=4096 originates from 4-bit color quantization in RGB channels ($4096=16^3$). In comparison, using 3-bit color for histogram binning leads to worse generation quality, and using 5-bit color yields little perceptual improvement.
>
> | OT solvers | Network Simplex (int) | Network Simplex (float) | Sinkhorn (float) |
> |-|:-:|:-:|:-:|
> | **Single-round OT solving latency (s) ↓** | 0.18 | 2.49 | 2.44 |
>
> In terms of OT solvers, entropy-regularized variants such as Sinkhorn provide speedups under floating-point cases. However, **our OT setting is naturally an integer programming problem, which can be solved more efficiently than the generic floating-point formulations targeted by alternative solvers**. We therefore use the network simplex method (integer formulation) by default.
>
> | Bin size (k) | k = 1 | k = 8 | k = 16 | k = 32 |
> |-|:-:|:-:|:-:|:-:|
> | **CLIP-Score ↑** | 25.91 | 27.49 | 29.25 | **29.41** |
> | **LAION-Aesthetics ↑** | 6.31 | 6.82 | **7.01** | 6.97 |
>
> We also perform ablations on the multi-option bin size k. Specifically, we mix RGB and RG quantization schemes to produce variants with different k values (with k=1 reducing to the single-option binning case). Overall, increasing k generally reduces perceptual deviation, though the improvement saturates around k=16 / k=32.
>
> ---
> > ### **Q: Choice of fixed norm (B) & the conversion between histogram and soft prompt**
>
> The main purpose of fixing B is to enable recovery of the soft prompt vector from its histogram representation (a normalized distribution) without requiring extra information. Empirically, we found that using **a smaller B (e.g., <10) leads to noticeably higher latency during soft prompt optimization. Other than that, B does not incur significant differences** (we let B = 40.0 by default).
>
> Meanwhile, we note that $||\ln(k h^{tgt})|| = B$ typically yields two solutions of k. In practice, the correct solution (corresponding to the proper normalizing factor) consistently falls within a reliable empirical interval, allowing us to efficiently locate it via a simple integer-level search and select the closest match.
>
> ---
> > ### **Q: User study on aesthetics**
>
> Per request, we conducted a user study with 12 participants to evaluate the aesthetics and naturalness of images generated by HIG versus StyleShot. For each (prompt, color scheme) pair, participants were shown outputs from both methods and asked to select the preferred image. Overall, participants chose HIG approximately 63% of the time, indicating that images generated by HIG are generally more visually appealing.

---

> ### Author Response · Authors · 2025-11-25
>
> > ### **Q: Optimal T selection**
>
> Based on the results in Appendix F (Table 5), we observe a clear trade-off between distribution alignment (HistKL) and generation quality (CLIP, Aesthetics). Empirically, **OT-based guidance applied in the early steps (i.e., closer to the Gaussian prior) has a limited control effect** due to the dominant noise. In contrast, **applying guidance in the middle or late denoising stages (e.g., the latter 50% of timesteps) is substantially more effective**. For simple constraints such as color-constrained generation, only 1-2 OT guidance steps are typically sufficient, whereas synthetic histograms derived from soft-prompt embeddings generally benefit from 3-4 steps to obtain noticeable quality improvements. We also find that evenly spaced guidance steps perform well in practice; **as long as guidance is applied within the appropriate temporal range, the guidance spacing has a negligible impact**.
>
> ---
> > ### **Q: Effects and details of latent histograms**
>
> Our intent in introducing latent histogram guidance was primarily to provide a **proof-of-concept** demonstrating that the HIG paradigm naturally generalizes to discrete latent token spaces, and the effect of control depends on the type of information encoded in the latent representation (e.g., colors, semantics, etc.). Unlike pixel histograms, performing explicit transformations on latent tokens involves far more confounding factors. In particular, **different latent spaces exhibit varying levels of robustness, leading to potentially heterogeneous control behaviors**. Empirically, applying OT-based transformations in more robust latent spaces tends to result in mild deviations from the original trajectory, while less robust spaces may yield noticeably coarser or even collapsed outputs. On the implementation side, we rely on pre-trained image tokenizers and their default vector-quantization metrics (most commonly L2 distance, with some models using cosine distance). Although task-specific fine-tuning could potentially make the latent space more stable and expressive, this direction is orthogonal to our focus and is not explored here.
>
> Our experiments in Fig. 8 and Appendix L aim to convey this core message: **performing HIG in latent space is feasible, and its effects are representation-dependent**. A full exploration of control outcomes under different latent spaces would require a dedicated study, and we view this as a promising direction for future work.

---

### Official Review · Reviewer_34Z3 · 2025-11-01

**Soundness:** 3
**Presentation:** 3
**Contribution:** 3
**Rating:** 4
**Confidence:** 4

**Summary:**

This paper introduces Histogram-Constrained Image Generation (HIG), a training-free, inference-time guidance technique for diffusion and flow-based generative models. It enforces user-specified statistical constraints, typically color or latent histogram, directly during sampling by applying optimal-transport based histogram matching to intermediate outputs. The algorithm alternates standard denoising steps with a histogram-projection step that minimally perturbs the diffusion trajectory while achieving the target distribution exactly. Experiments show precise histogram alignment and on-par realism and aesthetics.

**Strengths:**

**Mathematically sound.**
The proposed guidance method is mathematically grounded with an OT projection that ensures precise compliance with user-defined distributions.

**Training-free method.**
The proposed approach is training-free and can be integreated to any pretrained generator, which is shown in Fig. 9. This training-free feature implies wide application. It can also be applied to other domains where a distribution can be properly defined, such as 3D, video generation, etc.

**An interesting view of controllable image generation.**
This control mechanism serves as a novel paradigm, which extends controllable generation beyond textual and structural prompts to statistical constraints.

**Weaknesses:**

**Debatable histogram-constrained control paradigm.**
The core concept, imposing explicit histogram constraints during generation, may be philosophically and practically debatable. Perceptual style or appearance can be achieved more simply through existing style-transfer or color-transfer networks, which already approximate similar outcomes with lower computational cost. The mathematical precision of histogram alignment is appreciated, but this does not necessarily correspond to significant perceptual gains or artistic value, as shown in Tab. 1 (CLIP and Aesthetics), where improvements are marginal and debatable. Unfortunately, in real-world applications, users desire semantic or aesthetic control rather than strict statistical conformity.

**Limited exploration of latent-space histograms.**
Most experiments focus on color-space constraints. The analysis of latent histogram guidance remains preliminary in Fig. 8 and appendix L. The effect of changing histogram parameters on perceptual or semantic outcomes was not measured, as this would help understand the controllability of the proposed approach under latent guidance.

**Questions:**

1. Why is a solution to the OT problem corresponds to minimal impact on a trained latent generative model? Is this a pure hypothesis or can be mathematically proved?

2. How does computational cost scale with image resolution and number of bins? Since the OT solution is at least O(n^2) complexity, provided approximation is used, this would naturally scale poorly when generating ultra-high resolution images.

---

> ### Author Response · Authors · 2025-11-25
>
> Dear Reviewer 34Z3,
>
> We thank you for the time and effort in engaging with our work.
>
> ---
> > ### **Q: Benefit of histogram-constrained control paradigm**
>
> We wish to clarify that **HIG is not necessarily a user-centric control paradigm (e.g., ControlNets, LoRAs), and it is not intended as a drop-in replacement for classic tasks like style transfer**, where dedicated methods are available and may operate at lower computational cost. However, we want to highlight that **HIG's control paradigm does offer some nice properties that relevant baselines do not possess**.
>
> Because HIG applies explicit guidance transformations during inference, it provides two distinct capabilities:
> (1) **Exact value control**: HIG can enforce predetermined color sets or distributions with pixel-level fidelity, offering substantially stronger controllability than related baselines (as demonstrated in the color-constrained generation experiments through the HistKL metric);
> (2) **Strict distributional conformity**: The ability to match target histograms enables downstream applications where fidelity to prescribed statistics is essential, such as the information embedding demonstrations in the paper. Moreover, HIG’s control paradigm is also **flexible**: the same mechanism can be adapted to region-specific constraints (via masks) or to alternative histogram families with minimal modification. Overall, we wish to underscore that **HIG introduces a complementary and uniquely precise form of control for diffusion-based generative models**.
>
> ---
> > ### **Q: Exploration of latent histograms**
>
> Our intent in introducing latent histogram guidance was primarily to provide a **proof-of-concept** demonstrating that the HIG paradigm naturally generalizes to discrete latent token spaces, and the effect of control depends on the type of information encoded in the latent representation (e.g., colors, semantics, etc.). Unlike pixel histograms, performing explicit transformations on latent tokens involves far more confounding factors. In particular, **different latent spaces exhibit varying levels of robustness, leading to potentially heterogeneous control behaviors**. Empirically, applying OT-based transformations in more robust latent spaces tends to result in mild deviations from the original trajectory, while less robust spaces may yield noticeably coarser or even collapsed outputs.
>
> Our experiments in Fig. 8 and Appendix L aim to convey this core message: **performing HIG in latent space is feasible, and its effects are representation-dependent**. A full exploration of control outcomes under different latent spaces would require a dedicated study, and we view this as a promising direction for future work.
>
> ---
> > ### **Q: Whether OT-based transformation induces minimal impact**
>
> **A converged OT plan is guaranteed (by construction) to minimize the transport cost between the source and target distributions under the chosen metric (e.g., the L1 distance between color tuples).** This optimality can be mathematically established within the OT formulation. However, extending this guarantee to “perceptual deviation” is far more subtle: perceptual change does not admit a simple, tractable metric that aligns cleanly with OT’s cost function. Consequently, while the OT solution is provably optimal with respect to the selected distance metric, a minimal perceptual impact cannot be formally guaranteed.
>
> ---
> > ### **Q: Computation cost when scaling image resolution (HxW) and histogram dimension (d)**
>
> Regarding computational cost, we evaluate how the OT-solving latency scales with both image resolution and histogram dimension.
>
> | Image resolution (HxW) | 1024×1024 | 2048×2048 | 4096×4096 |
> |-|:-:|:-:|:-:|
> | **Single-round OT solving latency (s) ↓** | 0.18 | 0.17 | 0.21 |
>
> With fixed histogram dimension (d = 4096), **increasing image resolution does not introduce notable overhead in OT solving**.
>
> | Histogram dimension (d) | d = 512 (3-bit) | d = 4096 (4-bit) | d = 32768 (5-bit) |
> |-|:-:|:-:|:-:|
> | **CLIP-Score ↑** | 26.74 | 27.19 | 27.26 |
> | **LAION-Aesthetics ↑** | 6.59 | 6.78 | 6.81 |
> | **Single-round OT solving latency (s) ↓** | 0.01 | 0.18 | 10.47 |
>
> Under fixed resolution (1024x1024) and larger histogram dimension (d), **the OT computation becomes more expensive; however, we also found that enlarging d beyond the default config offers only marginal visual benefits**. FYI, our default choice d=4096 originates from 4-bit color quantization in RGB channels ($4096=16^3$). In comparison, using 3-bit color for histogram binning leads to worse generation quality, and using 5-bit color yields little perceptual improvement.

---

### Author Response · Authors · 2025-12-01
**Message to Area Chairs**

Dear Area Chairs,

We recognize that this year’s process presents unusual challenges due to the recent information leak incident, and we sincerely appreciate the ACs for taking on additional responsibilities during rebuttal. Thank you for your time and effort in evaluating our submission.

---

## **Method Summary**

Our work introduces **Histogram-Constrained Image Generation (HIG)**, a novel **distributional control paradigm** that enforces arbitrary histogram constraints during diffusion sampling. HIG is **training-free, efficient, and supports diverse applications**, such as constrained generation from color/latent-histogram and information embedding (i.e., hiding text messages within visually indistinguishable images).

---

## **Review Highlights: Strengths**

Reviewers 34Z3 notes that HIG represents “**a novel paradigm**” that “extends controllable generation beyond textual and structural prompts to statistical constraints”. The strengths of HIG are confirmed by multiple reviewers, including “**training-free**” (all), “**negligible inference overhead**” (ZzyM, 7m5F), “**wide application**” (34Z4, ZzyM), control with “**exact alignment**” (ZzyM, jNUJ), and “**compatibility with existing controls**” (ZzyM, jNUJ). Reviewer 7m5F offers strong praise for the **presentation**: “The paper is really well-presented. The generated images in the paper are visually appealing. Diagrams are clear.” Reviewer ZzyM leans towards acceptance (“I tend to accept the paper”), stating that “the paper presents a **technically sound framework** for histogram-constrained image generation via Optimal Transport.”

---

## **Review Highlights: Concerns & Replies**

> Two reviewers request clarifications on **the positioning of our method and its distinction/benefit compared to relevant tasks**: “Color-constrained generation appears conceptually similar to classical style transfer [...] The information embedding task also largely overlaps with diffusion-based steganography methods (jNUJ)”; “users desire semantic or aesthetic control rather than strict statistical conformity” (34Z3).

We wish to clarify that HIG is not primarily positioned as a user-centric control paradigm (e.g., ControlNets, LoRAs), and it is not intended as a drop-in replacement for classic tasks like style transfer, where dedicated methods are available and may operate at lower computational cost. Instead, **we position HIG as a generic distributional control paradigm, exemplified by three demo applications**. Our color-constrained experiments demonstrate that HIG offers **stronger color distribution control** than relevant baselines (such as style transfer); the information embedding technique aims to highlight HIG’s **control precision**; and the latent histogram examples illustrate HIG’s **generalization** to a broader family of histogram constraints.

In particular, HIG’s inference-time OT transformations enable **precise control** over both **the bin values** (e.g., exact colors or token indices) and **the distributional alignment** (i.e., the relative mass across bins), yielding fidelity that prior methods cannot reliably attain. Thus, **HIG should be viewed not as a substitute for specific tasks (e.g., style transfer, steganography), but as a complementary form of precise, distribution-level control**.

---

> ### Author Response · Authors · 2025-12-01
> **Message to Area Chairs (Cont'd)**
>
> ## **Review Highlights: Concerns & Replies (Cont'd)**
>
> > Two reviewers are interested in **the exploration of latent histograms**: “The analysis of latent histogram guidance remains preliminary in Fig. 8 and appendix L” (34Z3); “The perhaps more interesting case with latent histograms is less discussed” (7m5F).
>
> Our intent in introducing latent histogram guidance was primarily to provide a **proof-of-concept**, where we aim to communicate this core message: **performing HIG in latent space is feasible, and its effects are representation-dependent**. Unlike pixel histograms, performing explicit transformations on latent tokens involves far more confounding factors, and **different latent spaces exhibit varying levels of robustness, leading to potentially heterogeneous control behaviors**. Empirically, applying OT-based transformations in more robust latent spaces tends to result in mild deviations from the original trajectory. A full exploration of control outcomes under different latent spaces would require a dedicated study, and we view this as a promising direction for future work.
>
> > Two reviewers encourage **broader ablation and user studies**: Reviewer 34Z3 asks about the computational cost when scaling image resolution and number of bins; Reviewer ZzyM suggests clarifying the impact of histogram dimension (d), multi-option bin size (k), soft-prompt norm (B), guidance timestep (T), along with a user study on visual aesthetics.
>
> We have conducted the suggested studies. Specifically, we present quantitative evaluations for the major factors and empirical findings for the remaining ones. Please refer to our replies for the full results.
>
> ---
>
> ## **Additional Notes**
>
> **Reviewer jNUJ cites two works (‘DiffSteg’ [4] and ‘HiDiffusion’ [5]) that do not appear to exist**. Because these imaginary references form part of the reviewer’s novelty and comparison critique, we kindly ask that these portions of the review be appropriately reweighted in the final assessment.
>
> ---
>
> Thanks again for the time and effort you have given to our submission.
>
> Best regards,
>
> Authors of Submission15678

---

### Meta-Review · Area_Chair_QxRT · 2026-01-03

**Summary:**

The paper proposes Histogram-constrained Image Generation (HIG), which controls diffusion models by enforcing user-specified histogram-level constraints via optimal transport–based guidance during sampling. This intermediate-granularity control enables precise, interpretable, and flexible alignment with user intent while remaining compatible with existing control mechanisms. The major concerns are limited exploration of latent-space histograms, and unknown generalization and robustness. After reading the review and rebuttal, the meta-reviewer thinks the key concerns are valid and still outstanding, so would recommend rejection for this paper.

**Reviewer Concerns:**

Here are the major concerns that are still outstanding.

1.	Debatable histogram-constrained control paradigm (34Z3)
2.	Limited exploration of latent-space histograms (34Z3, 7m5F)
3.	Mathematical notations are unclear (ZzyM)
4.	Unknown generalization and robustness (ZzyM, jNUJ)
5.	The task might not be challenging enough (7m5F)
6.	Unclear application distinction (jNUJ)
7.	Limited novelty (jNUJ)

The following concerns are addressed.

1.	Incomplete empirical analysis (ZzyM)
2.	Unclear application distinction (jNUJ)

**Reviewer Scores:**

There is no response from the reviewers, I would assume they will all keep the scores as there are still key concerns unaddressed.

---

### Decision · Program_Chairs · 2026-01-26

Reject